# Normalized Rewards for Preference Optimization

**Shawn Im** [1]  **Federico Danieli** [2]  **Skyler Seto** [2]  **Barry-John Theobald** [2]  **Katherine Metcalf** [2]

## Abstract

Direct Alignment Algorithms (DAAs) such as DPO have become a common way to post-train and align LLMs with human preferences. However, DAAs have been observed to over-optimize their implicit reward model and decrease the likelihood of preferred responses. This results in a decrease in the total likelihood assigned to responses seen in the preference dataset, potentially resulting in undesirable behavior. To counteract this undesired side-effect of DAAs, we examine the effect of using objectives that add a regularization term to maintain the total length-normalized probabilities of the chosen and rejected responses. To better understand over-optimization, we investigate how response likelihood changes are distributed over the tokens with and without regularization. We find that a significant portion of the likelihood changes are due to a small set of outlier tokens, which explains how DAAs improve generation quality despite decreasing the likelihoods of chosen responses. We apply the proposed regularization to reference-based (DPO) and reference-free (SimPO) methods and find (1) improved trade-offs between generation quality and general benchmark capability and (2) improvements in reward modeling across datasets. For example, on Llama-3.1-8B-Instruct, we see both a $> 20\%$ relative increase in AlpacaEval2 scores and $> 9\%$ relative performance gains on general benchmarks. Additionally, we find that the added regularization term effectively mitigates the amount of displacement within preferred responses overall, and for the outlier tokens specifically, by utilizing low-likelihood tokens.

## 1. Introduction

With the rise in interactions between large language models (LLMs) and humans, training LLMs to produce responses that are considered desirable by human users has become a vital step. Such training is commonly performed with methods that learn what it means for a response to be desirable from a dataset of paired preferred and non-preferred responses, such as Reinforcement Learning from Human Feedback (RLHF) (Ouyang et al., 2022) and Direct Preference Optimization (DPO) (Rafailov et al., 2023). The main difference between the two approaches is RLHF relies on a two-step training process while DPO uses only one. RLHF first learns a reward function that assigns more value to the preferred versus non-preferred response, and then uses the reward function to train an LLM policy. DPO directly updates the LLM by maximizing the likelihood of the preferred versus non-preferred responses. Due to the simplicity and reduced computational cost of DPO, there has been a rise in the use and development of Direct Alignment Algorithms (DAAs), which train directly on the preferred and non-preferred paired dataset.

Despite their popularity, recent work has identified a critical limitation of DAAs: they can over-optimize their implicit reward model and ultimately constrain improvements to the quality of the LLM's generations (Rafailov et al., 2024; Razin et al., 2024; Huang et al., 2024). A common, concerning manifestation of over-optimization is likelihood displacement (Razin et al., 2024), a phenomenon whereby the likelihoods of both the preferred and the non-preferred responses drop simultaneously, potentially resulting in harmful behavior. However, despite these documented issues, DAAs are nonetheless still successfully used in post-training (Dubey et al., 2024; Groeneveld et al., 2024).

Several hypotheses have been proposed to explain the causes of over-optimization in DAAs: high embedding or textual similarity between preferred and non-preferred responses (Pal et al., 2024; Tajwar et al., 2024; Razin et al., 2024), or the insufficient regularization provided by the shape of the implicit reward function (Huang et al., 2024; Gupta et al., 2025). However, no single explanation satisfactorily generalizes across all DAA objectives. For example, a common difference between DAAs is the presence versus absence of a reference model in the reward com-

[1]UW-Madison [Work done at Apple] [2]Apple. Correspondence to: Shawn Im <sim29@wisc.edu>.

*Proceedings of the 43rd International Conference on Machine Learning*, Seoul, South Korea. PMLR 306, 2026. Copyright 2026 by the author(s).

putation (e.g., DPO uses a reference model for its reward, while SimPO (Meng et al., 2024) is reference-free), and hypotheses that explain over-optimization for a reference-free method do not hold or have not been applied to methods without a reference model and vice versa. This is evidenced by the fact that existing analyses on over-optimization (Yoon et al., 2025; Huang et al., 2024; Razin et al., 2024) focus on single instances of DAA, either reference-based or reference-free, but not both. The limitations of the current hypotheses motivate our research question: *how can we explain and mitigate reward over-optimization for both reference-based and reference-free rewards*?

In this work, we propose that reward over-optimization—particularly likelihood displacement—stems from a lack of normalization of the implicit reward. To counteract this, we introduce a regularization term designed to conserve the total response probability within preferred and non-preferred response pairs. To test the validity of our regularization term, we evaluate its impact on both reference-based (DPO) and reference-free (SimPO) methods. We find that its inclusion leads to (1) improved trade-offs between generation quality and general capability benchmarks, and (2) comparable or better reward modeling across datasets. Furthermore, our analysis reveals new insights into the mechanics of likelihood displacement. We discover that this phenomenon is highly concentrated, with a small subset of outlier tokens accounting for the majority of likelihood shift. Related observations of alignment primarily focusing on a small subset of tokens have been seen in works such as (Lin et al., 2023a; Qi et al., 2024). Building on a gradient analysis from Yoon et al. (2025), we find that low-likelihood tokens dominate the gradient, potentially causing the large shifts in outlier tokens, and our regularization term mitigates these shifts. These findings further support the use of our regularization term, and shed more light on why DAAs, like DPO and SimPO, improve generation despite causing likelihood displacement.

We provide a summary of the key results below:

1. We illustrate how the inability to fully account for the distribution of unseen responses leads to phenomena such as likelihood displacement.

2. We propose a mechanistic explanation of likelihood displacement, connecting its concentration on outlier tokens to perturbations of the partition function, and show via gradient analysis that our regularizer mitigates their outsized impact, with the corrective updates concentrated on low-likelihood tokens.

3. We demonstrate that our modified objective improves trade-offs between generation quality and benchmark performance (Lin et al., 2023b) for both DPO and SimPO.

## 2. Background

In this section, we introduce how RLHF and DPO utilize preference data.

**RLHF.** The preference learning component of RLHF is composed of two stages. The first consists of training a reward model $r_\phi(x, y)$ parameterized by $\phi$ on pairwise comparisons to assign scores to generated responses $y$ to a given prompt $x$. Given a pairwise preference $y_w \succ y_l$, the reward model $r_\phi(x, y)$ is trained to minimize the negative log-likelihood of the reward assignments under the Bradley-Terry model

$$-\log p(y_w \succ y_l) = -\log \sigma(r(x, y_w) - r(x, y_l)), \quad (1)$$

where $\sigma$ is the logistic function. Using this reward model in the second stage, the language model $\pi_\theta$ is then trained to maximize the expected reward of its responses under a KL constraint—added to mitigate drifting too far from the original model. The objective can be written as

$$\mathbb{E}_{x \sim D, y \sim \pi_\theta(\cdot|x)}\left[r(x, y) - \beta \mathbb{KL}[\pi_\theta(\cdot|x)||\pi_{\text{ref}}(\cdot|x)]\right], \quad (2)$$

where $\pi_\theta$ is the current model, $\pi_{\text{ref}}$ is the reference model, and $\beta$ is a hyperparameter for the KL constraint.

**DPO.** DPO is a direct alignment algorithm derived from the RLHF objective which removes the need for an external reward model and directly updates the language model using preference data. This is done by utilizing the fact that the optimal policy $\pi^*$ under the RLHF objective can be written as

$$\pi^*(y|x) = \frac{1}{Z(x)}\pi_{\text{ref}}(y|x)\exp\left(\frac{r(x, y)}{\beta}\right), \quad (3)$$

where $Z(x)$ is defined as

$$Z(x) = \sum_y \pi_{\text{ref}}(y|x)\exp\left(\frac{r(x, y)}{\beta}\right), \quad (4)$$

and denotes the partition function, which normalizes the output distribution of $\pi^*(y|x)$, ensuring the sum of response probabilities is 1. From this, we can write the reward $r(x, y)$ in terms of the optimal policy and the reference policy

$$r(x, y) = \beta \log \frac{\pi^*(y|x)}{\pi_{\text{ref}}(y|x)} + \beta \log Z(x). \quad (5)$$

Then, by defining

$$r_\theta(x, y) = \beta \log \frac{\pi_\theta(y|x)}{\pi_{\text{ref}}(y|x)} + \beta \log Z(x), \quad (6)$$

which corresponds to the reward model under which $\pi_\theta$ is optimal. We note that the partition function acts as an offset that ensures the rewards are calibrated and only when the

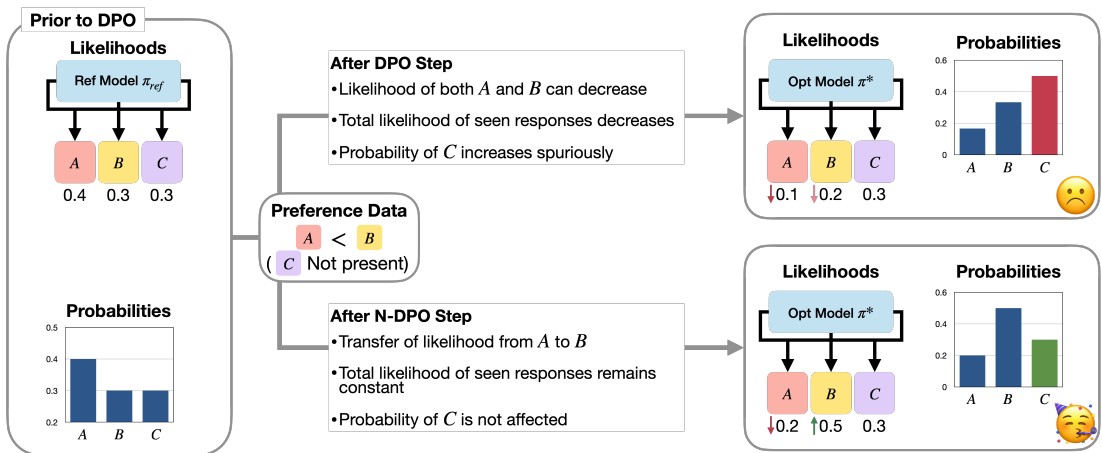

*Figure 1.* Illustrative example of likelihood displacement and how our regularization mitigates it. The reference model assigns likelihoods 0.4, 0.3, 0.3 to responses $A$, $B$, $C$, with matching probabilities since the partition function equals 1. We then train on the single preference $A < B$; the response $C$ is never observed. The example highlights the distinction between unnormalized likelihoods and normalized probabilities: under DPO, the likelihoods of both $A$ and $B$ decrease while $C$'s likelihood is unchanged, yet the resulting drop in the partition function spuriously inflates $C$'s probability. To counteract this, we introduce a regularization term that conserves the total likelihood over the seen pair $(A, B)$. The preference is still learned ($A$'s likelihood falls, $B$'s rises), but the partition function (and with it the probability of the unseen response $C$) is preserved.

partition function is 1 do the rewards directly correspond to log-likelihood ratios. By maximizing the likelihood of the policy under the Bradley-Terry model, the reward model and policy are simultaneously optimized, and under mild conditions, should result in the same optimal policy as RLHF. Expanding the reward terms, we have that the DPO objective $\mathcal{L}_{\text{DPO}}(\pi_\theta)$ is given by

$$\mathbb{E}_{(x,y_w,y_l)\sim\mathcal{D}} \left[ -\log \sigma \left( \beta \log \frac{\pi_\theta(y_w|x)\pi_{\text{ref}}(y_l|x)}{\pi_\theta(y_l|x)\pi_{\text{ref}}(y_w|x)} \right) \right],$$
(7)

with the partition function term cancelling out since the objective tracks the difference of rewards.

**SimPO.** A variant of DPO introduced to further simplify alignment training is SimPO. This variant removes the need for a reference model by using only the current model's likelihood of a response as the reward, and considers length-normalized probabilities. We can write the SimPO reward $r_\theta(x, y)$ for prompt $x$ and response $y$ as

$$r_\theta(x, y) = \beta \bar{\pi}_\theta(y|x) + Z(x),$$
(8)

where $\bar{\pi}_\theta(y|x) = \pi_\theta(y|x)^{1/|y|}$ is the length-normalized likelihood of a response for some model $\pi_\theta$ with $|y|$ being the length of the response and $Z(x)$ is the partition function defined as

$$Z(x) = \sum_y \pi_{\text{ref}}(y|x) \exp \left( \frac{r_\theta(x,y)^{|y|}}{\beta} \right).$$
(9)

In addition, SimPO introduces the use of a margin term to further separate the likelihoods of the preferred and non-preferred responses. Finally, the SimPO objective $\mathcal{L}_{\text{SimPO}}$

can be written as follows:

$$\mathbb{E}_{(x,y_w,y_l)\sim D} \left[ -\log \sigma \left( \beta \log \frac{\bar{\pi}_\theta(y_w|x)}{\bar{\pi}_\theta(y_l|x)} - \beta\gamma \right) \right], \quad (10)$$

where $\gamma$ is a hyperparameter for the margin size.

**Dangers of neglecting the partition function** Crucially, neither the DPO nor the SimPO objectives contain the partition function $Z(x)$. Indeed, since they consider the *differences* in rewards for each pair of responses, the $Z(x)$ terms cancel out in the respective objective formulations for DPO and SimPO, effectively rendering them invariant to changes in $Z(x)$. While mathematically convenient, this simplification deprives the model of an important factor — it does not incentivize the preservation of a good estimate of $Z(x)$. Furthermore, the limited set of responses available in the training set compared to the set of all possible responses makes proper normalization difficult. This difficulty in properly normalizing the rewards provides a compelling explanation for why likelihood displacement often occurs: there is insufficient information in the objective to learn calibrated likelihoods.

**Example** We provide an illustrative example of how this offset results in undesirable behavior in the trained model, shown in Figure 1. We start with a reference model that assigns a likelihood of 0.4 to option $A$ and 0.3 to options $B$ and $C$. Since we are considering a proper distribution, the partition function based on these likelihoods is 1. We can then consider training on a single preference $A < B$, where $C$ is not present. As outlined in (Razin et al., 2024), over-optimization might cause the likelihoods of $A$ and $B$

to drop simultaneously, which in turn causes the probability of the unseen response $C$ to spuriously increase. If instead we incentivize the conservation of the total likelihood between $A$ and $B$, we can still learn the preference correctly, while minimizing changes to responses outside the training distribution, preserving the probability of outputting the unseen response $C$. This allows the model to more reliably generate responses that more closely resemble the given preferences, and avoids generating unseen and potentially harmful responses.

## 3. Method

We first consider what the partition function for the rewards should be. If we consider any parameterized language model $\pi_\theta$ that applies softmax to its outputs, then the output distribution is always normalized and the partition function for the model is 1. As a result, we have the key property that *the partition function should remain fixed at 1 throughout training*.

However, the implicit DPO reward does not account for this constraint as the objective does not contain a term for the partition function resulting in phenomena such as likelihood displacement (Razin et al., 2024). Due to difficulties in effectively estimating the partition function for preference datasets (each prompt has only a single pair of responses), we propose a regularization term motivated by the insight of a fixed partition function while working around the challenge of accounting for unseen responses. We enforce normalization in the rewards by adding a regularization term that maintains the probability mass over the set of responses seen for a prompt. Given a preference data point with prompt $x$ and responses $y_w, y_l$, we start with a regularization penalty of

$$\lambda \left( \log \frac{\pi_\theta(y_w|x) + \pi_\theta(y_l|x)}{\pi_{\text{ref}}(y_w|x) + \pi_{\text{ref}}(y_l|x)} \right)^2, \qquad (11)$$

where $\lambda$ is a hyperparameter. The penalty aims to keep the ratio of the total response probability close to 1. Notice that the regularization term is minimized when the ratio of the total response probabilities is 1 and the total probability assigned to the two responses under the optimized model is the same as that under the original model. By maintaining the total probability, we mitigate offsets in likelihood that would occur given the lack of information about the full response distribution, and as a result, implicitly avoid large spurious changes. Unlike other regularization methods such as adding an SFT term on the preferred responses or DPO-P (Pal et al., 2024) that unconditionally increase the response likelihoods, our method aims to mitigate changes to unseen responses. However, response probabilities decrease exponentially with length and as responses are often hundreds of tokens long, the ratio of response probabilities

may be sensitive to differences in length or small changes in per-token probabilities. To have a more stable penalty and to mitigate length bias, we use length-normalized probabilities. Using $\bar{\pi}_\theta, \bar{\pi}_{\text{ref}}$ to denote length-normalized probabilities, we have the following regularization penalty:

$$\mathcal{R}(\pi_\theta, \pi_{\text{ref}}, x, y_w, y_l) = \lambda \left( \log \frac{\bar{\pi}_\theta(y_w|x) + \bar{\pi}_\theta(y_l|x)}{\bar{\pi}_{\text{ref}}(y_w|x) + \bar{\pi}_{\text{ref}}(y_l|x)} \right)^2. \qquad (12)$$

For consistency with the regularization, we modify the original objective with length-normalization which, with the regularization, gives the following objective for DPO:

$$\mathcal{L}_{\text{N-DPO}}(\theta) = \mathcal{L}_{\text{DPO}}(\pi_\theta) + \mathcal{R}(\pi_\theta, \pi_{\text{ref}}, x, y_w, y_l). \qquad (13)$$

We refer to the modified version of DPO as N-DPO. We also modify SimPO with the same form of regularization, but since SimPO is already length normalized, we simply add the regularization term resulting in:

$$\mathcal{L}_{\text{N-SimPO}}(\theta) = \mathcal{L}_{\text{SimPO}}(\pi_\theta) + \mathcal{R}(\pi_\theta, \pi_{\text{ref}}, x, y_w, y_l), \qquad (14)$$

which we refer to as N-SimPO.

## 4. Analysis

In this section, we consider an analysis of the effect of our regularization term by observing how it changes the likelihoods of responses over the course of training, at the level of the full response and at a token level. We track the dynamics of the response likelihoods over the course of training with and without our regularization. We then explore how the reward for each token changes by studying empirically the distribution of token-wise rewards with and without regularization. Through a theoretical gradient analysis, we reveal how likelihood displacement is distributed across tokens and how our regularization term uses low-likelihood tokens to reshape the reward distribution and response likelihoods.

### 4.1. Effect of Regularization

We analyze the dynamics of the response likelihoods for each of the methods on UltraFeedback (Cui et al., 2023) over the course of training, which we show in Figure 2. We demonstrate that adding regularization results in less likelihood displacement, particularly for DPO where the chosen responses maintain similar likelihood throughout training. In the case of SimPO, adding regularization keeps the chosen likelihoods similar while significantly decreasing the rejected response likelihoods, allowing the model to more strongly distinguish preferences without making the chosen response less likely. While we can see that the regularization does effectively mitigate displacement or result in stronger preference learning without worsening displacement, the response likelihoods do not explain where displacement

comes from and how exactly the regularization targets displacement.

To better understand displacement and the effect of regularization, we consider the overall token-wise reward distribution for each model and method as well as the distribution of the minimum token-wise reward per sample. The token-wise reward of a response $y$ is $\log \frac{\pi_\theta(y^{(i)})}{\pi_{\text{ref}}(y^{(i)})}$ where $y^{(i)}$ is the $i$th token. The minimum token-wise reward is $\min_{i \in [L]} \log \frac{\pi_\theta(y^{(i)}|x)}{\pi_{\text{ref}}(y^{(i)}|x)}$ where $L$ is the length of the response. To provide a clear comparison across settings, we use the change in log-likelihood per token as a normalized reward. We provide the results for Llama-3.1-8B-Instruct (Dubey et al., 2024) in Figure 3. When using DPO the peak of the minimum reward distribution is around -10 while the overall reward distribution lies mostly within -2.5 and 2.5, suggesting that for many samples when using DPO, the minimum reward lies far outside the typical range. This suggests that a significant part of likelihood displacement comes from outlier tokens significantly dropping the response likelihood. We demonstrate that this change does in fact reduce likelihood displacement by plotting the likelihood of responses over training in Figure 2. We also observe that while the overall token distribution does not change much between DPO and N-DPO, we see that there is a large shift in the minimum reward distribution. This suggests that N-DPO primarily mitigates the effect of these outlier tokens while maintaining the reward otherwise. We demonstrate how our regularization term mitigates these outlier tokens when total likelihood decreases as seen with Llama-3.1-8B-Instruct.

### 4.2. Token-wise Gradient Analysis

We consider the gradient of

$$\mathcal{R}(\pi_\theta, \pi_{\text{ref}}, x, y_w, y_l) = \lambda \left( \log \frac{\bar{\pi}_\theta(y_w|x) + \bar{\pi}_\theta(y_l|x)}{\bar{\pi}_{\text{ref}}(y_w|x) + \bar{\pi}_{\text{ref}}(y_l|x)} \right)^2 \tag{15}$$

with respect to $\theta$. First, we define the necessary notation. Let $P_\theta(x), P_{\text{ref}}(x)$ be the total length-normalized response probabilities under the trained and reference model respectively and let $\pi_\theta(y_{w/l}^{(i)})$ be the likelihood of the $i$th token in a response given the previous tokens. Then, we can write the gradient of the regularization term with respect to the parameters $\theta$ as

$$\nabla_\theta \mathcal{R} = \frac{2\lambda}{P_\theta(x)} \log \left( \frac{P_\theta(x)}{P_{\text{ref}}(x)} \right) \left( \frac{\bar{\pi}_\theta(y_w|x)}{|y_w|} \sum_{i=1}^{|y_w|} \frac{\nabla_\theta(\pi_\theta(y_w^{(i)}))}{\pi_\theta(y_w^{(i)})} \right.$$
$$\left. + \frac{\bar{\pi}_\theta(y_l|x)}{|y_l|} \sum_{i=1}^{|y_l|} \frac{\nabla_\theta(\pi_\theta(y_l^{(i)}))}{\pi_\theta(y_l^{(i)})} \right). \tag{16}$$

Notice that all of the token-wise contributions to the gradient have the same sign. The sign of the gradients are determined by $\log \left( \frac{P_\theta(x)}{P_{\text{ref}}(x)} \right)$ where if the total response probability has decreased, the gradients will be negative increasing token probabilities. If the total probability increases, the opposite will occur. Furthermore, looking at each token-wise gradient, we have $\frac{\nabla_\theta(\pi_\theta(y^{(i)}))}{\pi_\theta(y^{(i)})}$ which is inversely proportional to the likelihood of each token. If a token has small likelihood (e.g., $1e^{-7}$ times smaller than other tokens) the low-likelihood token's gradient will dominate. This has been observed in the gradient analysis in the ConfPO paper (Yoon et al., 2025). Then, if there is an outlier token with small likelihood and the response probability has decreased as seen with models such as Llama-3.1-8B-Instruct, the regularization term will strongly increase the likelihood of the outlier tokens. More generally, if the response likelihood has decreased significantly, the regularization term will prioritize updating the lowest likelihood tokens to increase the overall response likelihood. In this way, when the response likelihood decreases, the regularization term primarily shifts large negative rewards closer to 0.

We can also consider the case when the total response probability has increased. Now, the regularization term has gradients that will result in a decrease in response probability, but similar to before, these gradients will be dominated by the low likelihood tokens. Then, when total response probability has increased compared to the original, the regularization corrects this primarily by decreasing the likelihood of low likelihood tokens. In this way, the overall response probability is maintained with minimal changes to the most likely tokens, which are also most relevant for generation. In this way, the regularization term mitigates overly large likelihoods and does so with minimal distribution shift.

The gradient analysis reveals that introducing the regularization term effectively mitigates shifts in response likelihood and mitigates the presence of outlier tokens. Furthermore, we find that the regularization term does so primarily through low likelihood tokens which have a smaller effect on the overall sampling distribution. In this way, our regularization term demonstrates that low likelihood tokens not only provide an approximation of the gradient (Yoon et al., 2025) but also can be utilized to shape the reward and likelihood distribution.

## 5. Experiments

We evaluate the impact of accounting for reward normalization on downstream performance on instruction-following tasks, common sense and reasoning, and implicit reward modeling.

**Llama-3.1-8B-Instruct**

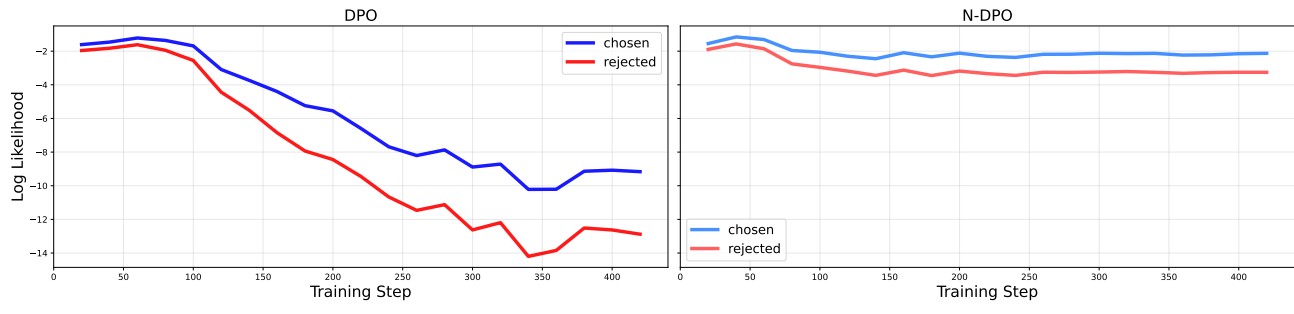

*(a)* Response likelihood over training with DPO vs. N-DPO

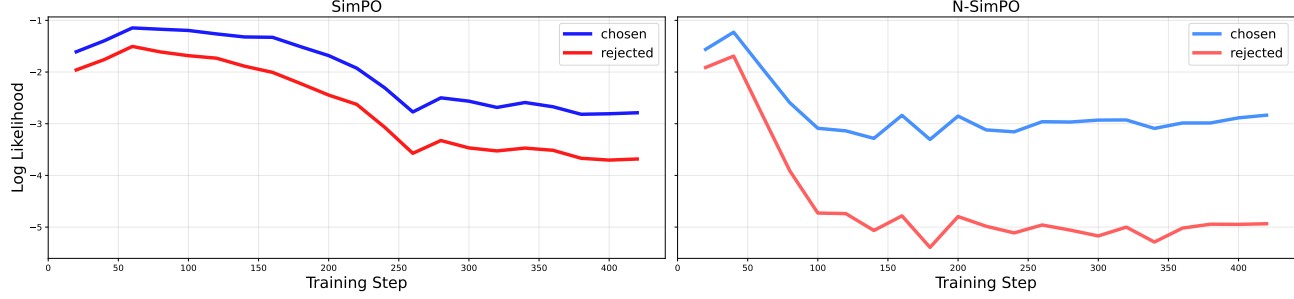

*(b)* Response likelihood over training with SimPO vs. N-SimPO

*Figure 2.* Comparison of response likelihoods between DPO/SimPO (left) and N-DPO/N-SimPO (right) for Llama-3.1-8B-Instruct. In blue is the chosen response likelihood and in red the rejected response likelihood.

### 5.1. Evaluation

We focus our evaluation on understanding the effect of our modifications by comparing DPO with N-DPO and SimPO with N-SimPO. We expect that due to the more strongly enforced normalization of rewards, likelihood displacement would be mitigated, and more generally, the model can learn from the preference data with less of a distribution shift. As a result, we expect to see better trade-offs between generation quality and benchmark performance when using N-DPO or N-SimPO. We also expect that with better normalized rewards, we may see better generalization of the implicit reward. We compare our methods to other forms of regularization such as DPO+SFT and DPOP (Pal et al., 2024).

We train Mistral-7B-Instruct (Jiang et al., 2023), Llama-3.1-8B-Instruct (Dubey et al., 2024), and OLMo-7B-SFT (Groeneveld et al., 2024) on Ultrafeedback (Cui et al., 2023) using DPO, N-DPO, SimPO, N-SimPO, DPO+SFT, and DPOP. For all runs, we train for 1 epoch with a cosine learning rate scheduler. We select hyperparameters using AlpacaEval1 results. We evaluate generation quality using AlpacaEval2 (Dubois et al., 2024) and assess the model on common sense and reasoning benchmarks (ARC, MMLU, HellaSwag, PIQA, SciQ, WinoGrande). We evaluate re-

ward modeling for a range of datasets (Ultrafeedback, HH-RLHF (Bai et al., 2022), HelpSteer2 (Wang et al., 2024)).

**Instruction Following.** AlpacaEval (Dubois et al., 2024) is a benchmark that evaluates a model based on its win-rate compared to GPT-3.5 (OpenAI, 2022) for AlpacaEval1 and GPT-4 (Achiam et al., 2023) for AlpacaEval2 using an LLM-as-a-judge in an instruction-following setting. For AlpacaEval2 both a raw win-rate (WR) and a length-controlled win-rate (LC) are provided. Table 1 shows the win-rates of the models on AlpacaEval1 and AlpacaEval2, where we can see that N-DPO and N-SimPO generally have improved performance over DPO and SimPO, respectively. In particular, we see over a 20% increase for AlpacaEval2 (LC) for Llama-3.1-8B-Instruct between DPO and N-DPO and a large increase in both AlpacaEval1 and AlpacaEval2 (WR) between SimPO and N-SimPO for OLMo-7B-SFT with over a 75% increase for AlpacaEval2 (WR). While we see a small decrease in instruction-following quality using N-SimPO for Llama-3.1-8B-Instruct, the benchmark performance noticeably improves. Furthermore, we find that N-DPO and N-SimPO perform better than both DPO+SFT and DPOP on AlpacaEval2, demonstrating the benefit of maintaining the total response probabilities rather than unconditionally increasing the likelihood of preferred responses.

*Table 1.* AlpacaEval scores across methods, with standard errors as subscripts. A1 corresponds to the AlpacaEval1 win-rate, while WR and LC correspond to the raw and length-controlled AlpacaEval2 win-rates, respectively. A1 results should be treated as validation metrics.

| | Mistral-7B-Instruct | | | Llama-3.1-8B-Instruct | | | OLMo-7B-SFT | | |
| --- | --- | --- | --- | --- | --- | --- | --- | --- | --- |
| | A1 | WR | LC | A1 | WR | LC | A1 | WR | LC |
| Reference | 93.17 | $14.54_{\pm1.14}$ | $18.17_{\pm0.49}$ | 90.00 | $25.09_{\pm1.39}$ | $18.84_{\pm0.17}$ | 58.15 | $2.92_{\pm0.54}$ | $4.50_{\pm0.28}$ |
| DPO | **94.66** | $18.13_{\pm1.24}$ | $23.66_{\pm0.49}$ | **91.67** | $24.18_{\pm1.36}$ | $21.72_{\pm0.27}$ | 79.63 | $8.12_{\pm0.89}$ | $8.96_{\pm0.34}$ |
| N-DPO | 94.28 | **$20.08_{\pm1.28}$** | **$25.48_{\pm0.50}$** | 91.28 | **$27.57_{\pm1.43}$** | **$24.97_{\pm0.20}$** | 81.24 | **$11.79_{\pm1.04}$** | **$9.88_{\pm0.29}$** |
| SimPO | 90.87 | $19.68_{\pm1.28}$ | **$26.34_{\pm0.42}$** | **88.93** | **$28.68_{\pm1.46}$** | **$26.74_{\pm0.12}$** | 71.21 | $3.66_{\pm0.61}$ | $5.15_{\pm0.28}$ |
| N-SimPO | **92.72** | **$20.38_{\pm1.30}$** | $26.05_{\pm0.45}$ | 85.45 | $27.11_{\pm1.43}$ | $25.54_{\pm0.18}$ | **79.63** | **$10.02_{\pm0.96}$** | $8.70_{\pm0.31}$ |
| DPO+SFT | 92.98 | $18.32_{\pm1.24}$ | $20.11_{\pm0.53}$ | 76.43 | $4.07_{\pm0.63}$ | $4.12_{\pm0.19}$ | 66.83 | $3.70_{\pm0.61}$ | $2.69_{\pm0.15}$ |
| DPOP | 94.52 | $14.76_{\pm1.13}$ | $18.22_{\pm0.48}$ | 91.28 | $20.73_{\pm1.28}$ | $15.92_{\pm0.17}$ | 70.50 | $3.86_{\pm0.62}$ | $4.42_{\pm0.27}$ |

**Llama-3.1-8B-Instruct**

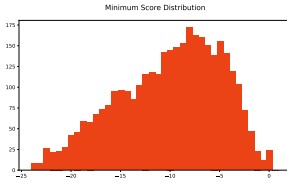

*(a)* Minimum of token likelihood changes per sample for DPO

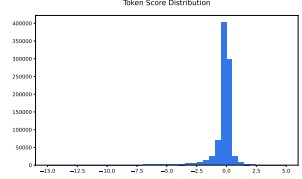

*(b)* Overall distribution of token likelihood changes for DPO

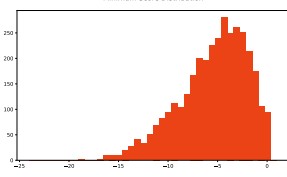

*(c)* Minimum of token likelihood changes per sample for N-DPO

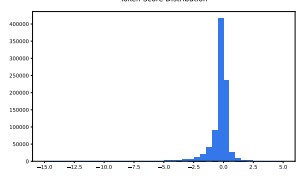

*(d)* Overall distribution of token likelihood changes for N-DPO

*Figure 3.* Comparison of token-wise reward distributions between DPO and N-DPO for Llama-3.1-8B-Instruct. On the left is the distribution of the minimum token reward per sample, and on the right is the distribution of all token rewards.

**Common Sense and Reasoning.** In addition to instruction-following quality, we perform evaluation on various benchmarks from LM Evaluation Harness (Gao et al., 2024) to see how well the model maintains its general common sense and reasoning capabilities. We expect reducing likelihood displacement to also reduce distribution shift allowing for better benchmark performance. To quantify how well the model maintains benchmark performance, we consider the difference in scores between the reference model and the fine-tuned model. The results for the evaluation are shown in Table 2. We can see that for Llama-3.1-8B-Instruct, N-DPO and N-SimPO result not only in better maintenance of benchmark

performance, but also improve scores on average by over 3%. For Mistral-7B-Instruct, we also see an improvement between DPO versus N-DPO along with a small drop in benchmark performance that is accompanied by an increase in generation quality by a larger margin than the drop in benchmark performance. For OLMo-7B-SFT, we find that N-DPO improves benchmark performance while improving generation quality. We note that for N-SimPO, there is a drop in benchmark performance, but the results suggest this is due to difficulties with applying SimPO to OLMo: generation quality increased the most with the smallest learning rate, margin, and beta. This suggests that larger updates using SimPO do not benefit OLMo-7B-SFT, and that N-SimPO creates a better optimization landscape for OLMo-7B-SFT, allowing for more dramatic changes in weaker base models that improve generation quality. Compared to DPO+SFT and DPOP, across models, we see that N-DPO and N-SimPO have comparable benchmark performance. We provide the hyperparameter sweep range and final hyperparameters in Appendix B.

**Implicit Reward.** We further evaluate the trained models on their implicit reward modeling capabilities on both Ultra-Feedback and datasets not seen during training (HH-RLHF, HelpSteer2). Table 3 shows the reward accuracy on the eval splits for these datasets using a length-normalized reward. We find that DPO+SFT and DPOP have a larger drop in reward accuracy compared to other methods.

### 5.2. Regularization Ablation

We perform an ablation on the regularization term by setting $\lambda = 0$ to demonstrate that maintaining the probability mass helps improve performance beyond using only length-normalization (L-DPO). We provide the results on AlpacaEval in Appendix A, where we can see that performance improves when $\lambda$ is a non-zero value. In Appendix A we additionally report the sensitivity of reward accuracy and reward margins to the choice of $\lambda$, confirming that N-DPO

*Table 2.* Common sense and reasoning benchmarks performance across methods, with standard errors as subscripts. DPOP uses the best setting for each model.

**Mistral-7B-Instruct**

| | MMLU | ARC Chal | ARC Easy | HellaSwag | PiQA | SciQ | WinoG | Avg |
|---|---|---|---|---|---|---|---|---|
| Reference | $44.32_{\pm0.41}$ | $49.49_{\pm1.46}$ | $70.33_{\pm0.94}$ | $62.86_{\pm0.48}$ | $75.19_{\pm1.01}$ | $90.90_{\pm0.91}$ | $61.48_{\pm1.37}$ | $64.94$ |
| DPO | $-1.34_{\pm0.41}$ | $-3.41_{\pm1.46}$ | $-5.05_{\pm0.98}$ | $-1.45_{\pm0.49}$ | $-2.67_{\pm1.04}$ | $-3.00_{\pm1.03}$ | $-2.68_{\pm1.38}$ | $-2.80$ |
| N-DPO | $\mathbf{+0.17}_{\pm0.41}$ | $\mathbf{-1.11}_{\pm1.46}$ | $\mathbf{-1.90}_{\pm0.95}$ | $\mathbf{-0.95}_{\pm0.48}$ | $\mathbf{-1.96}_{\pm1.03}$ | $\mathbf{-0.80}_{\pm0.94}$ | $\mathbf{-0.31}_{\pm1.37}$ | $-0.98$ |
| SimPO | $-0.70_{\pm0.41}$ | $-3.84_{\pm1.46}$ | $-3.12_{\pm0.96}$ | $-7.47_{\pm0.50}$ | $-5.06_{\pm1.07}$ | $-0.50_{\pm0.93}$ | $-2.21_{\pm1.38}$ | $-3.27$ |
| N-SimPO | $-0.84_{\pm0.41}$ | $-4.10_{\pm1.46}$ | $-3.62_{\pm0.97}$ | $-6.87_{\pm0.49}$ | $-4.79_{\pm1.06}$ | $-1.20_{\pm0.96}$ | $-2.44_{\pm1.38}$ | $-3.41$ |
| DPO+SFT | $\mathbf{+2.57}_{\pm0.29}$ | $\mathbf{+6.14}_{\pm1.45}$ | $\mathbf{+4.75}_{\pm0.89}$ | $\mathbf{+2.34}_{\pm0.48}$ | $\mathbf{+3.37}_{\pm0.96}$ | $\mathbf{+1.60}_{\pm0.83}$ | $\mathbf{+6.40}_{\pm1.31}$ | $\mathbf{+3.88}$ |
| DPOP | $+1.65_{\pm0.29}$ | $+0.25_{\pm1.46}$ | $+0.80_{\pm0.93}$ | $+1.38_{\pm0.48}$ | $+0.44_{\pm1.00}$ | $+0.30_{\pm0.90}$ | $+0.87_{\pm1.36}$ | $+0.81$ |

**Llama-3.1-8B-Instruct**

| | MMLU | ARC Chal | ARC Easy | HellaSwag | PiQA | SciQ | WinoG | Avg |
|---|---|---|---|---|---|---|---|---|
| Reference | $43.28_{\pm0.41}$ | $52.82_{\pm1.46}$ | $81.44_{\pm0.80}$ | $57.52_{\pm0.49}$ | $79.76_{\pm0.94}$ | $95.20_{\pm0.68}$ | $67.40_{\pm1.32}$ | $68.20$ |
| DPO | $\mathbf{+4.71}_{\pm0.41}$ | $-4.78_{\pm1.46}$ | $-9.51_{\pm0.92}$ | $\mathbf{+6.43}_{\pm0.48}$ | $-5.82_{\pm1.02}$ | $-4.20_{\pm0.91}$ | $-9.55_{\pm1.39}$ | $-3.25$ |
| N-DPO | $+4.24_{\pm0.41}$ | $\mathbf{+3.84}_{\pm1.45}$ | $\mathbf{+0.80}_{\pm0.78}$ | $+6.24_{\pm0.48}$ | $\mathbf{+0.82}_{\pm0.92}$ | $\mathbf{+0.20}_{\pm0.66}$ | $\mathbf{+5.13}_{\pm1.25}$ | $\mathbf{+3.04}$ |
| SimPO | $+1.69_{\pm0.41}$ | $+4.77_{\pm1.44}$ | $+0.17_{\pm0.80}$ | $-5.49_{\pm0.50}$ | $-0.54_{\pm0.82}$ | $+0.80_{\pm0.62}$ | $\mathbf{+6.55}_{\pm1.23}$ | $+1.14$ |
| N-SimPO | $\mathbf{+8.49}_{\pm0.41}$ | $\mathbf{+9.21}_{\pm1.42}$ | $\mathbf{+2.27}_{\pm0.76}$ | $-0.92_{\pm0.49}$ | $\mathbf{+1.36}_{\pm0.91}$ | $\mathbf{+1.10}_{\pm0.60}$ | $+4.33_{\pm1.26}$ | $\mathbf{+3.69}$ |
| DPO+SFT | $+5.61_{\pm0.29}$ | $-4.61_{\pm1.46}$ | $-0.59_{\pm0.81}$ | $-1.65_{\pm0.50}$ | $+0.27_{\pm0.93}$ | $+0.40_{\pm0.65}$ | $+4.27_{\pm1.27}$ | $+0.53$ |
| DPOP | $+7.08_{\pm0.29}$ | $+0.41_{\pm1.46}$ | $+0.59_{\pm0.79}$ | $+0.36_{\pm0.49}$ | $+0.22_{\pm0.93}$ | $+0.00_{\pm0.68}$ | $+1.03_{\pm1.31}$ | $+1.38$ |

**OLMo-7B-SFT**

| | MMLU | ARC Chal | ARC Easy | HellaSwag | PiQA | SciQ | WinoG | Avg |
|---|---|---|---|---|---|---|---|---|
| Reference | $37.94_{\pm0.40}$ | $39.33_{\pm1.43}$ | $67.97_{\pm0.96}$ | $53.54_{\pm0.50}$ | $76.66_{\pm0.99}$ | $91.10_{\pm0.90}$ | $63.93_{\pm1.35}$ | $61.50$ |
| DPO | $-0.15_{\pm0.40}$ | $-0.85_{\pm1.42}$ | $-2.06_{\pm0.97}$ | $+0.27_{\pm0.50}$ | $-1.42_{\pm1.01}$ | $-4.80_{\pm1.09}$ | $-2.05_{\pm1.36}$ | $-1.58$ |
| N-DPO | $\mathbf{+0.47}_{\pm0.40}$ | $\mathbf{+1.88}_{\pm1.44}$ | $\mathbf{+2.11}_{\pm0.94}$ | $\mathbf{+3.52}_{\pm0.49}$ | $-0.82_{\pm1.00}$ | $-1.20_{\pm0.95}$ | $-0.95_{\pm1.36}$ | $\mathbf{+0.72}$ |
| SimPO | $-0.04_{\pm0.40}$ | $\mathbf{-0.34}_{\pm1.43}$ | $\mathbf{-0.25}_{\pm0.96}$ | $\mathbf{-1.17}_{\pm0.50}$ | $\mathbf{-1.09}_{\pm1.02}$ | $\mathbf{-0.70}_{\pm0.93}$ | $\mathbf{-1.50}_{\pm1.36}$ | $\mathbf{-0.73}$ |
| N-SimPO | $\mathbf{+0.10}_{\pm0.40}$ | $-1.53_{\pm1.42}$ | $-14.01_{\pm0.98}$ | $-3.49_{\pm0.50}$ | $-4.79_{\pm1.05}$ | $-4.20_{\pm1.07}$ | $-5.76_{\pm1.39}$ | $-4.81$ |
| DPO+SFT | $-0.62_{\pm0.28}$ | $-4.18_{\pm1.40}$ | $-8.21_{\pm1.01}$ | $-3.46_{\pm0.50}$ | $-0.49_{\pm0.99}$ | $-12.80_{\pm1.30}$ | $-4.50_{\pm1.38}$ | $-4.89$ |
| DPOP | $\mathbf{+1.30}_{\pm0.28}$ | $+0.00_{\pm1.42}$ | $\mathbf{+2.23}_{\pm0.94}$ | $\mathbf{+2.34}_{\pm0.50}$ | $\mathbf{+2.23}_{\pm0.96}$ | $-1.20_{\pm0.94}$ | $\mathbf{+2.05}_{\pm1.34}$ | $\mathbf{+1.28}$ |

*Table 3.* Reward accuracy on UltraFeedback (UF), HH-RLHF (HH), and HelpSteer2 (HS) across methods.

| | Mistral-7B-Instruct | | | Llama-3.1-8B-Instruct | | | OLMo-7B-SFT | | |
|---|---|---|---|---|---|---|---|---|---|
| | UF | HH | HS | UF | HH | HS | UF | HH | HS |
| DPO | 80.73 | 55.12 | **66.15** | 74.71 | **59.55** | 68.23 | 71.12 | 53.30 | 64.06 |
| N-DPO | **94.28** | **81.94** | 56.33 | **77.20** | 58.73 | **75.00** | **73.26** | **54.63** | 64.32 |
| SimPO | **82.12** | **57.17** | **64.32** | **76.22** | 57.03 | 67.19 | 60.19 | 55.47 | 53.65 |
| N-SimPO | 82.06 | 56.82 | 63.28 | 75.81 | **57.62** | 67.97 | **65.05** | **55.68** | **56.77** |
| DPO+SFT | 53.18 | 50.96 | 59.11 | 37.67 | 46.86 | 32.03 | 59.14 | 51.91 | **66.15** |
| DPOP | 72.74 | 56.74 | 55.73 | 61.92 | 59.28 | 55.21 | 67.01 | 54.90 | 55.99 |

remains stable across the swept range.

# 6. Related Works

**DAA Methods.** A wide range of works have explored various approaches to improve existing DAA methods like DPO (Rafailov et al., 2023). One class of changes is modi-fying the function of the reward margin (Azar et al., 2024; Zhao et al., 2023; Tang et al., 2024). A range of works have approached the problem of over-optimization and likelihood displacement through modifying the reward function based on alternatives to KL regularization (Huang et al., 2024; Gupta et al., 2025; Wang et al., 2023), focusing on select tokens (Yoon et al., 2025), mitigating length-normalization

exploits (Gupta et al., b), distilling explicit reward models (Fisch et al., 2024), or adding regularization (Liu et al., 2024; Pal et al., 2024). Another line of work has proposed modifications to the DPO objective for robustness such as rDPO (Chowdhury et al., 2024) and ROPO (Liang et al., 2024), and other works have explored weighting samples (Zhou et al., 2024) or using rejection sampling (Xiong et al., 2023; Zhao et al., 2024; Liu et al., 2023). A range of works have considered various forms of data for aligning language models using reward data (Chen et al., 2024a) or data from self play (Wu et al., 2024; Gupta et al., a; Tang et al., 2025). Other objectives include KTO (Ethayarajh et al., 2024), which uses prospect theory to motivate an objective that considers a set of desirable responses and undesirable responses, and ORPO (Hong et al., 2024), which utilizes the log odds ratio between the preferred and dispreferred responses for optimization.

**Reward over-optimization.** Outside of proposing new DAAs to mitigate over-optimization, works have also analyzed factors that may lead to over-optimization and likelihood displacement (Rafailov et al., 2024). Razin et al. (2024) studies under a simplified model how the embedding geometry may lead to likelihood displacement. Pal et al. (2024) provides an analysis of how likelihood displacement can arise given preference pairs with small edit distance. Reward over-optimization is also a generally observed phenomenon outside of DAAs with RLHF (Chen et al., 2024b) and other reinforcement learning settings (Skalse et al., 2022; Ibarz et al., 2018).

## 7. Discussion

Our results suggest that a common factor in reward over-optimization and likelihood displacement across methods and, in particular, across both reference-based and reference-free methods is the lack of reward normalization. The objectives modified to normalize rewards, N-DPO and N-SimPO, demonstrate better trade-offs between generation quality and benchmark performance, sometimes improving both while also maintaining reward modeling abilities. We note that these modifications require tuning an additional hyper-parameter, although our sensitivity analysis in Appendix A shows that N-DPO is robust to choices of $\lambda$. The improvement across various axes suggests that reward normalization has a significant role in DAAs and that enforcing such constraints can be an effective addition to methods. Furthermore, our analysis of token-wise rewards demonstrates that likelihood displacement does not affect the model broadly, but rather primarily on a limited number of tokens, explaining why generation improves despite decreasing likelihood. A token-wise analysis of the gradient of our proposed regularization term demonstrates that our regularization term effectively mitigates such outlier tokens and more generally

utilizes low likelihood tokens to reshape the reward and likelihood distribution, see Appendix A.1 and A.2. These analyses provide insight into the role of different tokens in preference optimization and demonstrate the need for finer-grained analyses of reward model behavior. While our analysis and regularization term do not generalize to online settings or methods such as KTO (Ethayarajh et al., 2024), we hope our analysis can motivate more general analyses and methodologies.

## Impact Statement

Post-training with methods such as DPO and other DAAs are essential for allowing interactions between people and models to be beneficial and safe. Our study of likelihood displacement and regularization aims to allow for more robust learning from human preferences and to better understand how post-training affects models.

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

# A. Additional Results

We provide results with and without regularization for length-normalized DPO.

*Table 4.* AlpacaEval scores for length-normalized DPO with and without regularization. WR corresponds to the raw win-rate and LC corresponds to the length-controlled win-rate.

**Mistral-7B-Instruct**

|  | AlpacaEval1 | AlpacaEval2 (WR) | AlpacaEval2 (LC) |
|---|---|---|---|
| $\lambda = 0$ | 93.03 | 16.54 | 20.21 |
| N-DPO | 94.28 | 18.16 | 22.10 |

**Llama-3.1-8B-Instruct**

|  | AlpacaEval1 | AlpacaEval2 (WR) | AlpacaEval2 (LC) |
|---|---|---|---|
| $\lambda = 0$ | 91.25 | 30.56 | 28.50 |
| N-DPO | 91.28 | 31.84 | 29.41 |

**OLMo-7B-SFT**

|  | AlpacaEval1 | AlpacaEval2 (WR) | AlpacaEval2 (LC) |
|---|---|---|---|
| $\lambda = 0$ | 78.43 | 8.76 | 7.34 |
| N-DPO | 81.24 | 9.21 | 8.02 |

We provide AlpacaEval scores for the DPO+SFT method in Table 5.

*Table 5.* AlpacaEval scores (win-rate) across methods.

|  | **Mistral-7B-Instruct** | **Llama-3.1-8B-Instruct** | **OLMo-7B-SFT** |
|---|---|---|---|
|  | AlpacaEval1 | AlpacaEval1 | AlpacaEval1 |
| Reference | 93.17 | 90.00 | 58.15 |
| DPO | **94.66** | 91.67 | 79.63 |
| N-DPO | 94.28 | 91.28 | **81.24** |
| DPO+SFT | 92.15 | **93.52** | 66.50 |

We report the sensitivity of reward accuracy and reward margins to the regularization strength $\lambda$ in Tables 6 and 7.

*Table 6.* Sensitivity of reward accuracy to the regularization strength $\lambda$. Each cell reports the reward accuracy of the best hyperparameter configuration (selected by AlpacaEval1 win-rate) at the corresponding $\lambda$, from a joint sweep over $\beta$, $\gamma$, and learning rate.

|  | MISTRAL-7B-INSTRUCT | | LLAMA-3.1-8B-INSTRUCT | | OLMo-7B-SFT | |
|---|---|---|---|---|---|---|
| $\lambda$ | N-DPO | N-SIMPO | N-DPO | N-SIMPO | N-DPO | N-SIMPO |
| 0.025 | 84.61 | 82.18 | 79.05 | 78.59 | 75.98 | 65.22 |
| 0.05 | 84.20 | 82.06 | 79.17 | 78.94 | 75.87 | 63.25 |
| 0.1 | 83.28 | 81.77 | 78.94 | 79.34 | 76.16 | 62.62 |

*Table 7.* Sensitivity of reward margins to the regularization strength $\lambda$, under the same evaluation setup as Table 6. Margins decrease monotonically with $\lambda$ for both methods, as expected: stronger regularization limits how far the model can separate chosen from rejected responses.

| $\lambda$ | MISTRAL-7B-INSTRUCT | | LLAMA-3.1-8B-INSTRUCT | | OLMO-7B-SFT | |
|---|---|---|---|---|---|---|
| | N-DPO | N-SIMPO | N-DPO | N-SIMPO | N-DPO | N-SIMPO |
| 0.025 | 2.355 | 7.223 | 3.301 | 6.143 | 2.588 | 0.318 |
| 0.05 | 2.238 | 7.084 | 3.221 | 5.784 | 2.444 | 0.268 |
| 0.1 | 2.092 | 6.856 | 3.050 | 5.439 | 2.299 | 0.213 |

Outlier token comparisons for DPO vs. N-DPO, and SimPO vs. N-SimPO for both Mistral-7B-Instruct-v0.2 and OLMo-7B-SFT-hf.

## A.1. Mistral-7B-Instruct-v0.2

*Table 8.* Outlier category distribution: DPO vs N-DPO for Mistral-7B-Instruct-v0.2.

| Category | DPO | | N-DPO | | Change |
|---|---|---|---|---|---|
| | Count | % | Count | % | ($\Delta$%) |
| function_word | 542 | 18.8 | 570 | 19.4 | +0.6 |
| number | 69 | 2.4 | 67 | 2.3 | -0.1 |
| other_single_char | 0 | 0.0 | 3 | 0.1 | +0.1 |
| punctuation | 351 | 12.2 | 337 | 11.5 | -0.7 |
| single_letter | 157 | 5.5 | 147 | 5.0 | -0.4 |
| special_token | 46 | 1.6 | 36 | 1.2 | -0.4 |
| subword_fragment | 1541 | 53.5 | 1549 | 52.8 | -0.7 |
| symbol | 18 | 0.6 | 21 | 0.7 | +0.1 |
| whitespace | 156 | 5.4 | 203 | 6.9 | +1.5 |

*Table 9.* Outlier summary: DPO vs N-DPO or Mistral-7B-Instruct-v0.2.

| Statistic | DPO | N-DPO |
|---|---|---|
| Total tokens | 58,655 | 58,655 |
| Num outliers | 2,880 | 2,933 |
| Outlier % | 4.9 | 5.0 |
| Threshold | -2.0000 | -3.3151 |
| Reward mean | -0.2955 | -0.4805 |
| Reward std | 1.0574 | 1.5650 |
| Outlier reward mean | -4.0886 | -6.0234 |
| Outlier reward min | -22.6446 | -40.2568 |

*Table 10.* Shared and unique outlier tokens: DPO vs N-DPO for Mistral-7B-Instruct-v0.2.

| Token | Category | Count |
|---|---|---|
| *Shared outliers* | | |
| . | punctuation | 135 |
| <newline> | whitespace | 117 |
| the | function_word | 77 |
| , | punctuation | 60 |
| a | single_letter | 53 |
| and | function_word | 50 |
|  | special_token | 46 |
| in | function_word | 35 |
| is | function_word | 33 |
| <whitespace> | whitespace | 31 |
| *DPO only* | | |
| powerful | subword_fragment | 3 |
| h | single_letter | 3 |
| art | subword_fragment | 2 |
| actually | subword_fragment | 1 |
| personality | subword_fragment | 1 |
| Because | subword_fragment | 1 |
| Sil | subword_fragment | 1 |
| `, | subword_fragment | 1 |
| par | subword_fragment | 1 |
| gle | subword_fragment | 1 |
| *N-DPO only* | | |
| depend | subword_fragment | 2 |
| technology | subword_fragment | 2 |
| things | subword_fragment | 1 |
| equations | subword_fragment | 1 |
| rot | subword_fragment | 1 |
| accessible | subword_fragment | 1 |
| custom | subword_fragment | 1 |
| stay | subword_fragment | 1 |
| ]); | punctuation | 1 |
| List | subword_fragment | 1 |

*Table 11.* Outlier category distribution: SimPO vs N-SimPO for Mistral-7B-Instruct-v0.2.

| Category | SimPO | | N-SimPO | | Change |
|---|---|---|---|---|---|
| | Count | % | Count | % | (Δ%) |
| function_word | 728 | 24.8 | 685 | 23.4 | -1.5 |
| number | 30 | 1.0 | 36 | 1.2 | +0.2 |
| punctuation | 440 | 15.0 | 420 | 14.3 | -0.7 |
| single_letter | 166 | 5.7 | 168 | 5.7 | +0.1 |
| special_token | 13 | 0.4 | 16 | 0.6 | +0.1 |
| subword_fragment | 1364 | 46.5 | 1425 | 48.6 | +2.1 |
| symbol | 13 | 0.4 | 13 | 0.4 | +0.0 |
| whitespace | 179 | 6.1 | 170 | 5.8 | -0.3 |

*Table 12.* Outlier summary: SimPO vs N-SimPO for Mistral-7B-Instruct-v0.2.

| Statistic | SimPO | N-SimPO |
|---|---|---|
| Total tokens | 58,655 | 58,655 |
| Num outliers | 2,933 | 2,933 |
| Outlier % | 5.0 | 5.0 |
| Threshold | -6.1987 | -5.4903 |
| Reward mean | -0.9325 | -0.8133 |
| Reward std | 2.4994 | 2.2270 |
| Outlier reward mean | -9.5135 | -8.5701 |
| Outlier reward min | -31.5423 | -32.0055 |

*Table 13.* Shared and unique outlier tokens: SimPO vs N-SimPO for Mistral-7B-Instruct-v0.2

| Token | Category | Count |
|---|---|---|
| *Shared outliers* | | |
| . | punctuation | 199 |
| <newline> | whitespace | 147 |
| the | function_word | 118 |
| , | punctuation | 95 |
| a | single_letter | 90 |
| and | function_word | 87 |
| is | function_word | 47 |
| that | function_word | 41 |
| to | function_word | 38 |
| in | function_word | 36 |
| *SimPO only* | | |
| By | function_word | 2 |
| over | subword_fragment | 2 |
| knows | subword_fragment | 1 |
| approach | subword_fragment | 1 |
| wages | subword_fragment | 1 |
| stopping | subword_fragment | 1 |
| application | subword_fragment | 1 |
| latest | subword_fragment | 1 |
| touch | subword_fragment | 1 |
| first | subword_fragment | 1 |
| *N-SimPO only* | | |
| certain | subword_fragment | 2 |
| Because | subword_fragment | 1 |
| ", | punctuation | 1 |
| seems | subword_fragment | 1 |
| em | subword_fragment | 1 |
| follow | subword_fragment | 1 |
| b | single_letter | 1 |
| Che | subword_fragment | 1 |
| Who | subword_fragment | 1 |
| distinguish | subword_fragment | 1 |

## A.2. OLMo-7B-SFT-hf

*Table 14.* Outlier category distribution: DPO vs N-DPO for OLMo-7B-SFT-hf.

| Category | DPO | | N-DPO | | Change |
| --- | --- | --- | --- | --- | --- |
| | Count | % | Count | % | (Δ%) |
| content_word | 362 | 35.6 | 1339 | 48.4 | +12.8 |
| function_word | 139 | 13.7 | 410 | 14.8 | +1.1 |
| number | 38 | 3.7 | 75 | 2.7 | -1.0 |
| other_single_char | 3 | 0.3 | 6 | 0.2 | -0.1 |
| punctuation | 70 | 6.9 | 178 | 6.4 | -0.5 |
| single_letter | 54 | 5.3 | 120 | 4.3 | -1.0 |
| special_token | 37 | 3.6 | 58 | 2.1 | -1.5 |
| subword_fragment | 92 | 9.1 | 260 | 9.4 | +0.3 |
| symbol | 21 | 2.1 | 69 | 2.5 | +0.4 |
| whitespace | 200 | 19.7 | 249 | 9.0 | -10.7 |

*Table 15.* Outlier summary: DPO vs N-DPO for OLMo-7B-SFT-hf.

| Statistic | DPO | N-DPO |
| --- | --- | --- |
| Total tokens | 55,261 | 55,261 |
| Num outliers | 1,016 | 2,764 |
| Outlier % | 1.8 | 5.0 |
| Threshold | -2.0000 | -2.4403 |
| Reward mean | -0.1585 | -0.3777 |
| Reward std | 1.3822 | 1.3455 |
| Outlier reward mean | -5.9049 | -4.5450 |
| Outlier reward min | -57.3944 | -27.8938 |

*Table 16.* Shared and unique outlier tokens: DPO vs N-DPO for OLMo-7B-SFT-hf.

| Token | Category | Count |
|---|---|---|
| *Shared outliers* | | |
| `<newline>` | whitespace | 197 |
| `<|endoftext|>` | special_token | 37 |
| `.` | punctuation | 24 |
| `<space>I` | single_letter | 18 |
| `<space>the` | function_word | 13 |
| `U+FFFD` | symbol | 12 |
| `<space>is` | function_word | 11 |
| `<space>and` | function_word | 10 |
| `1` | number | 8 |
| ``` ``` ``` | subword_fragment | 8 |
| *DPO only* | | |
| `<space>85` | number | 3 |
| `'d` | subword_fragment | 2 |
| `Current` | subword_fragment | 1 |
| `<space>p` | single_letter | 1 |
| `Por` | subword_fragment | 1 |
| `<space>generate` | content_word | 1 |
| `<space>Part` | content_word | 1 |
| `<space>until` | content_word | 1 |
| `<space>Fellow` | content_word | 1 |
| `<space>confidence` | content_word | 1 |
| *N-DPO only* | | |
| `<space>we` | content_word | 12 |
| `<space>have` | function_word | 12 |
| `<space>more` | content_word | 8 |
| `<space>By` | function_word | 7 |
| `<space>know` | content_word | 6 |
| `<space>help` | content_word | 5 |
| `<space>them` | content_word | 5 |
| `3` | number | 5 |
| `Cell` | subword_fragment | 5 |
| `magnetic` | subword_fragment | 5 |

*Table 17.* Outlier category distribution: SimPO vs N-SimPO for OLMo-7B-SFT-hf.

| Category | SimPO | | N-SimPO | | Change |
|---|---|---|---|---|---|
| | Count | % | Count | % | (Δ%) |
| content_word | 149 | 33.9 | 204 | 47.8 | +13.9 |
| function_word | 46 | 10.4 | 75 | 17.6 | +7.1 |
| number | 12 | 2.7 | 13 | 3.0 | +0.3 |
| punctuation | 24 | 5.5 | 36 | 8.4 | +3.0 |
| single_letter | 46 | 10.4 | 49 | 11.5 | +1.0 |
| special_token | 10 | 2.3 | 9 | 2.1 | -0.2 |
| subword_fragment | 13 | 3.0 | 17 | 4.0 | +1.0 |
| symbol | 1 | 0.2 | 2 | 0.5 | +0.2 |
| whitespace | 139 | 31.6 | 22 | 5.2 | -26.4 |

*Table 18.* Outlier summary: SimPO vs N-SimPO for OLMo-7B-SFT-hf.

| Statistic | SimPO | N-SimPO |
|---|---|---|
| Total tokens | 55,261 | 55,261 |
| Num outliers | 440 | 427 |
| Outlier % | 0.8 | 0.8 |
| Threshold | -2.0000 | -2.0000 |
| Reward mean | -0.0609 | -0.0576 |
| Reward std | 0.5155 | 0.4505 |
| Outlier reward mean | -4.1299 | -3.2282 |
| Outlier reward min | -15.0571 | -9.6681 |

*Table 19.* Shared and unique outlier tokens: SimPO vs N-SimPO for OLMo-7B-SFT-hf.

| Token | Category | Count |
|---|---|---|
| *Shared outliers* | | |
| `<newline>` | whitespace | 139 |
| `<space>I` | single_letter | 41 |
| `<|endoftext|>` | special_token | 10 |
| `.` | punctuation | 9 |
| `<space>you` | content_word | 8 |
| `!` | punctuation | 7 |
| `<space>and` | function_word | 7 |
| `<space>here` | content_word | 7 |
| `'m` | subword_fragment | 6 |
| `2` | number | 6 |
| *SimPO only* | | |
| `<space>How` | content_word | 1 |
| `<space>Part` | content_word | 1 |
| *N-SimPO only* | | |
| `<space>is` | function_word | 3 |
| `<space>based` | content_word | 2 |
| `<space>not` | function_word | 2 |
| `<space>write` | content_word | 1 |
| `<space>Great` | content_word | 1 |
| `<space>may` | function_word | 1 |
| `<space>certainly` | content_word | 1 |
| `<space>models` | content_word | 1 |
| `<space>services` | content_word | 1 |
| `30` | number | 1 |

# B. Hyperparameters

We provide the set of hyperparameters used to perform hyperparameter sweeps for each method and model.

|  | **Mistral-7B-Instruct** | **Llama-3.1-8B-Instruct** | **OLMo-7B-SFT** |
|---|---|---|---|
| DPO | [0.03, 0.1, 0.3] | [0.01, 0.03, 0.1] | [0.03, 0.1, 0.3] |
| N-DPO | [0.3, 1.0, 3.0] | [0.3, 1.0, 3.0] | [0.3, 1.0, 3.0] |
| SimPO | [0.3, 1.0, 3.0] | [0.3, 1.0, 3.0] | [0.3, 1.0, 3.0] |
| N-SimPO | [3.0] | [3.0] | [0.3] |

*Table 20.* Set of $\beta$ used for hyperparameter sweep for each model/method

|  | **Mistral-7B-Instruct** | **Llama-3.1-8B-Instruct** | **OLMo-7B-SFT** |
|---|---|---|---|
| SimPO | [0.4, 0.8, 1.2, 1.6, 2.0] | [0.4, 0.8, 1.2, 1.6, 2.0] | [0.4, 0.8, 1.2, 1.6, 2.0] |
| N-SimPO | [0.8, 1.2, 1.6, 2.0] | [0.8, 1.2, 1.6, 2.0] | [0.8, 1.2, 1.6, 2.0] |

*Table 21.* Set of $\gamma$ used for hyperparameter sweep for each model/method

|  | **Mistral-7B-Instruct** | **Llama-3.1-8B-Instruct** | **OLMo-7B-SFT** |
|---|---|---|---|
| N-DPO | [0.0, 0.025, 0.05, 0.075, 0.1] | [0.0, 0.025, 0.05, 0.075, 0.1] | [0.0, 0.025, 0.05, 0.075, 0.1] |
| N-SimPO | [0.025, 0.05, 0.075, 0.1] | [0.025, 0.05, 0.075, 0.1] | [0.025, 0.05, 0.075, 0.1] |

*Table 22.* Set of $\lambda$ used for hyperparameter sweep for each model/method

| **Mistral-7B-Instruct** | **Llama-3.1-8B-Instruct** | **OLMo-7B-SFT** |
|---|---|---|
| $[3e-8, 1e-7, 3e-7]$ | $[1e-7, 3e-7, 1e-6]$ | $[1e-7, 3e-7, 1e-6]$ |

*Table 23.* Set of learning rates used for hyperparameter sweep for each model

|  | **Mistral-7B-Instruct** | **Llama-3.1-8B-Instruct** | **OLMo-7B-SFT** |
|---|---|---|---|
| $\beta$ | (0.03/1.0/3.0/3.0) | (0.01/3.0/3.0/3.0) | (0.03/3.0/0.3/0.3) |
| $\gamma$ | (0/0/1.6/2.0) | (0/0/0.8/2.0) | (0/0/0.4/1.6) |
| $\lambda$ | (0/0.1/0/0.1) | (0/0.025/0/0.025) | (0/0.05/0/0.025) |
| Learning Rate | (1e-7/3e-7/1e-7/1e-7) | (3e-7/1e-6/3e-7/1e-6) | (3e-7/1e-6/1e-7/3e-7) |

*Table 24.* Hyperparameters used for model evaluation. (DPO/N-DPO/SimPO/N-SimPO)

# C. Gradient Derivation

We provide a derivation of the gradient of the regularization term

$$\mathcal{R}(\pi_\theta, \pi_{\text{ref}}, x, y_w, y_l) = \lambda \left( \log \frac{\bar{\pi}_\theta(y_w|x) + \bar{\pi}_\theta(y_l|x)}{\bar{\pi}_{\text{ref}}(y_w|x) + \bar{\pi}_{\text{ref}}(y_l|x)} \right)^2 \tag{17}$$

with respect to parameters $\theta$. Letting $g(\pi_\theta) = \left( \log \frac{\bar{\pi}_\theta(y_w|x) + \bar{\pi}_\theta(y_l|x)}{\bar{\pi}_{\text{ref}}(y_w|x) + \bar{\pi}_{\text{ref}}(y_l|x)} \right)$, we have that

$$\nabla_\theta \mathcal{R} = 2\lambda g(\pi_\theta) \nabla_\theta g(\pi_\theta) \tag{18}$$

Now, defining $h(\pi_\theta) = \frac{\bar{\pi}_\theta(y_w|x) + \bar{\pi}_\theta(y_l|x)}{\bar{\pi}_{\text{ref}}(y_w|x) + \bar{\pi}_{\text{ref}}(y_l|x)}$ and denoting the denominator of $h(x)$ as $P_{\text{ref}}(x)$, we have

$$\nabla_\theta g(\pi_\theta) = \frac{1}{h(\pi_\theta)} \nabla_\theta h(\pi_\theta) = \frac{P_{\text{ref}}(x)}{\bar{\pi}_\theta(y_w|x) + \bar{\pi}_\theta(y_l|x)} \frac{1}{P_{\text{ref}}(x)} \nabla_\theta(\bar{\pi}_\theta(y_w|x) + \bar{\pi}_\theta(y_l|x)) \tag{19}$$

Writing $\bar{\pi}_\theta(y|x)$ as $\pi_\theta(y|x)^{1/|y|}$ and decomposing $\pi_\theta(y|x)^{1/|y|}$ as

$$\exp \left( \frac{1}{|y|} \sum_{i=1}^{|y|} \log \pi_\theta(y^{(i)}) \right) \tag{20}$$

we have

$$\nabla_\theta(\bar{\pi}_\theta(y_w|x) + \bar{\pi}_\theta(y_l|x)) = \left( \frac{\bar{\pi}_\theta(y_w|x)}{|y_w|} \sum_{i=1}^{|y_w|} \frac{\nabla_\theta \pi_\theta(y_w^{(i)})}{\pi_\theta(y_w^{(i)})} + \frac{\bar{\pi}_\theta(y_l|x)}{|y_l|} \sum_{i=1}^{|y_l|} \frac{\nabla_\theta \pi_\theta(y_l^{(i)})}{\pi_\theta(y_l^{(i)})} \right) \tag{21}$$

Through the chain rule and letting $P_\theta(x) = \bar{\pi}_\theta(y_w|x) + \bar{\pi}_\theta(y_l|x)$, we have that

$$\nabla_\theta \mathcal{R} = \frac{2\lambda}{P_\theta(x)} \log \left( \frac{P_\theta(x)}{P_{\text{ref}}(x)} \right) \left( \frac{\bar{\pi}_\theta(y_w|x)}{|y_w|} \sum_{i=1}^{|y_w|} \frac{\nabla_\theta(\pi_\theta(y_w^{(i)}))}{\pi_\theta(y_w^{(i)})} + \frac{\bar{\pi}_\theta(y_l|x)}{|y_l|} \sum_{i=1}^{|y_l|} \frac{\nabla_\theta(\pi_\theta(y_l^{(i)}))}{\pi_\theta(y_l^{(i)})} \right). \tag{22}$$

