# OpenReview forum: "Normalized Rewards for Preference Optimization"
_ICML.cc/2026/Conference — ICML 2026 regular_

### Official Review · Reviewer_JH42 · 2026-03-09

**Soundness:** 3
**Presentation:** 3
**Significance:** 2
**Originality:** 2
**Overall Recommendation:** 4
**Confidence:** 4

**Summary:**

The paper proposes adding a regularizer to direct alignment algorithms, more particularly DPO and SimPO. The regularizer is intended to mitigate a problem identified in previous work, namely likelihood displacement, where given a positive and negative pair, the probability of both decreases. More specifically, the regularizer tries to incentivize the optimization to preserve the sum of the likelihood of the positive relative to the negative examples with respect to the base/reference model ( i.e. if one goes up another must go down ) . The results show little improvement in generation quality, as the difference relative to the non-regularized version is small and below the combined error. However, when evaluated on a suite of common sense and reasoning benchmarks, the paper shows that, compared with DPO/SimPO, the regularized version loses less performance on these benchmarks and in some cases even improves. There are also significant gains in reward modeling quality.

**Compliance With Llm Reviewing Policy:**

Affirmed.

**Final Justification:**

The rebuttal acknowledged my main concerns and improved my confidence in the revision, but it did not change my overall evaluation.

**Key Questions For Authors:**

What were the validation datasets? You have to explain this in the text.

In Figure 2, it appears that only N-DPO addresses the likelihood displacement issue, not N-simPO. Does this mean that the regularizer does not mitigate the likelihood displacement issue in simPO?

It would be good to align the Y-scales of the plots in Figure 2, which use the same method, so we can more clearly see the relative magnitude difference between positive and negative values.

**Limitations:**

yes

**Strengths And Weaknesses:**

## weaknesses

- **The experimental evidence for improved generation quality is weak.** The reported gains are extremely small, and several of the comparisons appear to be of the same order as the error bars ( i.e. the delta is below the combined standard error). In cases where improvements are this marginal, the standard error  information should not just be put into a separate table in appendix, but instead should be made directly visible in the main figure through error bars or explicit ± vales. As currently presented, the results section gives a stronger impression of improvement than the underlying evidence supports. I think this is important to be discussed in the text as well.


- **The broader framing around the point that only a few tokens have their token likelihood changes should also be made more careful with respect to prior work.** The paper seems to suggest that the observation that alignment only meaningfully affects a relatively small subset of token probabilities is a newly identified phenomenon for likelihood displacement. However, this is more a property of aligned models more generally and has been studied before. It has long been understood that alignment/post-training often changes only limited parts of the model’s output distribution rather than uniformly reshaping behavior everywhere. Early works include [Lin et al 2023], but I am sure there are more.


- **The paper also does not clearly explain how hyperparameters were selected.** In particular, it is not stated which dataset or datasets were used for tuning, nor whether the authors followed a clean validation protocol that kept the test benchmarks fully isolated from model selection. The paper should specify the validation data used, and make clear that no test-set results were used to guide configuration decisions.


[Lin et al 2023] THE UNLOCKING SPELL ON BASE LLMS:
RETHINKING ALIGNMENT VIA IN-CONTEXT LEARNING

## Strengths

- The proposed regularizer is simple and well motivated, and it connects naturally to heuristics that prior work has used to address displacement, such as only training on a subset of tokens. As outline, the regularized does a similar effect  more naturally through the gradient of the regularizer itself, which makes the method easier to interpret and grounded in a relatively simple and understandable principle.

- The preservation of performance relative to the reference model on common sense and reasoning benchmarks is a meaningful strength of the paper and suggests that the approach has practical utility. Even if the gains in generation quality are limited, showing that the regularized variant better retains or sometimes even improves performance on these benchmarks is valuable, since it points to a more favorable tradeoff between the new capability added and the previous ones the model had already.

- The explanation of the method is clear, and the overall motivation and analysis are also presented clearly.

---

> ### Author Rebuttal · Authors · 2026-03-31
>
> **Gains in generation quality are "extremely small"**
>
> We respectfully push back on this characterization. Consider the key results on Llama-3.1-8B-Instruct:
> - AlpacaEval2 WR: DPO 27.70 → N-DPO **31.84** (a gain of 4.14 points, well outside the individual standard errors of ~1.3)
> - AlpacaEval2 LC: DPO 24.46 → N-DPO **29.41** (a gain of ~5 points)
>
> We acknowledge that gains are more modest in other settings — for instance, Mistral-7B SimPO → N-SimPO shows WR 17.61 → 18.87 and LC 23.28 → 23.67. At the same time, we note that AlpacaEval's relatively small sample size (805 examples) inherently limits how tight the error bars can be. While the overall picture is clearly model-dependent, the improvements on Llama-3.1-8B remain substantial. Still, we agree standard errors should be included in the main tables to make it easier to assess statistical significance across all comparisons: we thank the reviewer for the recommendation, and will adjust this in the revised paper.
>
> **Novelty of outlier token analysis**
>
> We appreciate the reference provided by the reviewer (which we will include) and agree that the observation that alignment affects a limited subset of token probabilities is not new. We acknowledge that our paper gives the impression of claiming this observation as novel, particularly in the phrasing of contribution 2 (lines 93-95). We will reword this in the revised paper to make clear that our contribution is not the observation itself, but rather stems from the *explanation* we hypothesize for this phenomenon and the *solution* we propose to counteract it. Indeed, we connect the concentration of likelihood displacement on outlier tokens to perturbations of the partition function induced by the alignment procedure, and propose a principled regularizer to counteract it, whose effectiveness on said tokens is justified by the gradient analysis in Sec4.2.
>
> **Hyperparameter selection/validation protocol**
>
> The reviewer raises a very important point. While in Sec5.1 we describe the training setup and in AppB we provide hyperparameter sweep ranges, the paper in its current state fails to clearly define the selection criteria used to choose the final configuration. To clarify: we selected hyperparameters based on AlpacaEval1 win-rate, and all other evaluations (AE2, common sense and reasoning benchmarks, reward modeling) were not used during model selection. We will make this protocol explicit in the revised paper, including a note that the AE1 results reported in Tab1 should be interpreted as validation rather than test metrics.
>
> **Improving Fig2**
>
> We agree that aligning the y-scales and combining chosen/rejected likelihoods on the same plot would make the figure much clearer, and we will revise accordingly. Regarding the different behavior of N-DPO and N-SimPO, rather than eliminating displacement, in the latter case the regularizer appears to enable stronger preference learning — N-SimPO achieves a larger reward margin (wider gap between chosen and rejected likelihoods) with a similar degree of displacement. We believe this explains the improved downstream performance, though we acknowledge the effect is less clean than for N-DPO.

---

> > ### Author Rebuttal · Reviewer_JH42 · 2026-04-01
> >
> > I thank the authors for mitigating my concerns, especially regarding the evaluation methodology. I agree that “extremely small” was too strong a characterization. That said, my overall assessment remains unchanged as my broader view about the paper’s overall contribution and empirical strength still stands.

---

> > > ### Author Response · Authors · 2026-04-08
> > >
> > > We thank the reviewer for the fair reassessment of the generation quality gains, and for the constructive feedback throughout: we believe the revised paper will be stronger for it.
> > >
> > > We provide the extended baseline comparison (DPOP and DPO+SFT) in our response to Reviewer 2jMo, with discussion in our response to Reviewer C1x4. We also include a token-level outlier analysis (response to Reviewer C1x4, point 2) showing that the regularizer reduces outlier severity, as predicted by the gradient analysis. All other promised revisions (std's in main tables, Fig2 improvements, clarifying validation protocol) remain planned for the final version.

---

### Official Review · Reviewer_on7F · 2026-03-10

**Soundness:** 3
**Presentation:** 3
**Significance:** 3
**Originality:** 2
**Overall Recommendation:** 4
**Confidence:** 4

**Summary:**

The paper addresses the "likelihood displacement" problem in Direct Alignment Algorithms (DAAs). The authors argue that since DPO and SimPO ignore the partition function $Z(x)$, the resulting reward model is uncalibrated. They propose N-DPO and N-SimPO, which incorporate a regularization term to stabilize the total probability mass of response pairs. While the empirical results on Llama-3.1-8B are quite high, there are several concerns regarding the technical depth and the rigor of the presentation.

**Compliance With Llm Reviewing Policy:**

Affirmed.

**Key Questions For Authors:**

See weakness

**Limitations:**

yes

**Strengths And Weaknesses:**

Strengths:
1. The paper addresses a well-known theoretical gap in DPO—the missing partition function. The explanation of why this causes likelihood displacement is easy to follow.
2. The discovery that outlier tokens drive the drop in likelihood is very interesting. The gradient analysis explains how the new term specifically helps these tokens.
3. The method (N-DPO and N-SimPO) shows clear gains on AlpacaEval and general benchmarks (like MMLU and ARC).

Weakness:
1. The new term introduces a new hyperparameter $\lambda$. Looking at the appendix, the best value for $\lambda$ changes depending on the model and the method. This might make the training process more complex for users.
2. For the OLMo model, N-SimPO actually saw a drop in general benchmark scores. While the authors discuss this briefly, it suggests the method might not be as stable on smaller or weaker base models.
3. The paper mentions other ways to stop over-optimization (like adding an SFT loss). A more detailed comparison with these existing tricks in the main text would help show exactly how much better this specific approach is.

---

> ### Author Rebuttal · Authors · 2026-03-31
>
> **Lambda hyperparameter adds complexity**
>
> We agree with the reviewer’s concern. We will explicitly flag the added tuning burden and include results for the whole lambda sweep in the revised paper: these are reported in our response to Reviewer C1x4. While the optimal lambda is model-dependent, we point out that a small sweep over [0, 0.1] was sufficient across all models, and that the ablation in Sec5.2 confirms that the regularization consistently helps over the lambda=0 baseline.
>
> **N-SimPO drops benchmark scores on OLMo**
>
> As we discuss in the paper, SimPO on OLMo required very conservative hyperparameters (smallest learning rate, margin, and beta among all models), suggesting a delicate optimization landscape for this model. N-SimPO appears to enable more aggressive preference optimization, which improves generation quality but comes at the cost of reasoning benchmark performance. This suggests that for weaker base models, additional care in hyperparameter selection may be needed. We will expand this discussion in the revised paper.
>
> **More detailed comparison with SFT regularization**
>
> We thank the reviewer for the recommendation. We will expand on the results and clarify key differences, mainly that SFT regularization increases chosen likelihoods unconditionally, while ours aims to prevent redistribution of likelihood to unseen responses. See also our response to Reviewer 2jMo for a detailed discussion.

---

> > ### Author Rebuttal · Reviewer_on7F · 2026-04-05
> >
> > Thanks for your response and I will be maintaining my score.

---

> > > ### Author Response · Authors · 2026-04-08
> > >
> > > We provide the full DPO-P and DPO+SFT comparison in our response to Reviewer 2jMo (Tables 1-3), with detailed discussion in our response to Reviewer C1x4. The results confirm that our N-DPO approach remains competitive or superior across all three evaluation axes (generation quality, common sense benchmarks, and reward accuracy), even when compared against methods specifically designed to prevent likelihood collapse. We hope the additional baselines and the expanded discussion of SFT regularization vs our approach satisfactorily address the reviewer's remaining concerns, and would be grateful if the reviewer could reconsider their score in light of the new evidence.

---

### Official Review · Reviewer_C1x4 · 2026-03-12

**Soundness:** 3
**Presentation:** 3
**Significance:** 2
**Originality:** 2
**Overall Recommendation:** 4
**Confidence:** 4

**Summary:**

The paper aim to fix the issue of decreasing likelihood of preferred responses, over the course of training, in DAAs, by adding a regularizer that maintains the total length normalized probability of the rejected and chosen responses. They additionaly show that this change in likelihood is significantly accounted for, by a small set of outlier tokens. Experimentally, they show that their proposed approach, leads to improvements in generation / discrimination quality alongside improvements in a set of benchmarks evaluations.

**Compliance With Llm Reviewing Policy:**

Affirmed.

**Final Justification:**

I upgraded my score to 4. The rebuttal addresses my questions and N-DPO seems to improve over DPO-Positive in terms of reward accuracy while the alpacaeval and LM harness results are other way around, The new results would meaningfully improve the paper.

My only concern is the mixed results across models, as also noted by other reviewers

**Key Questions For Authors:**

See Weaknesses

**Limitations:**

Yes

**Strengths And Weaknesses:**

## Strengths
- The paper is well written and easy to follow
- The authors clearly motivate the necessity of a regularizer, to constrain probability mass of chosen and rejected responses, and experimentally show their approach leads to improvements over Vanilla DPO/SimPO. The proposed approach is simple and easy to implement.

## Weaknesses
- The authors do not provide any confidence intervals, after running their experiments on multiple seeds. Its hard to make statements about improvements wrt baseline, without any confidence intervals, given the results are close to that of Vanilla DPO/SimPO across most cases.
- Missing citations/related work. The authors do not cite or mention [1], which is the first work to study about Reward Overoptimization in DAAs, and also studies the problem of decreasing likelihood of preferred responses in a toy MDP setting. Another related work is [2].
- Limited number of baselines in the experiments. The problem of decreasing likelihood of preferred responses in DAAs has been relatively well studied. The authors should compare their proposed approach with other approaches that target the same problem, such as [2],[3].
- The paper mentions outlier tokens contribute significantly to the change in likelihood. Real examples illustrating what these tokens are would greatly help in making this point clear. Furthermore, a generation case-study comparing the model generations of their proposed approach vs the baselines is also not provided.
- Ablations. The authors provide a regularization ablation in Section 5.2, but this still doesn't investigate the sensitivity of their proposed approach to the regularization parameter $\lambda$.
- Minor Issues. It would help if the chosen and rejected responses were in the same plot in figure 2, so that the claims in section 4 are more easily verified.

[1] Scaling Laws for Reward Model Overoptimization in Direct Alignment Algorithms, Rafailov et al

[2] Robust Preference Optimization through Reward Model Distillation, Fisch et al

[3] Smaug: Fixing Failure Modes of Preference Optimisation with DPO-Positive, Pal et al

---

> ### Author Rebuttal · Authors · 2026-03-31
>
> **Confidence intervals**
>
> We point out that standard errors are already included in AppA, but we agree that the paper's clarity would benefit from reporting them directly in the main tables, and will address this in the revised version. See also our response to Reviewer JH42 for a more detailed discussion.
>
> **Outlier tokens analysis**
>
> While we do not currently have a detailed analysis characterizing the outlier tokens, as a preliminary observation we noticed outlier tokens are frequently punctuation marks or eos tokens. Regarding a generation case study, we note that the paper's core contribution is at the level of reward normalization and token-wise dynamics rather than generation-level differences, so we prioritize the token-level analysis as more directly informative for the paper's claims.
>
> **Limited baselines comparisons**
>
> We thank the reviewer for pointing out the relevant related work. We focused our comparison on DPO+SFT (see Tab5) as it is the most established representative of regularization methods which prevent likelihood collapse, and we will expand on those results in the final version (see also our response to Reviewer 2jMo). We appreciate the reference to [2], but while it does address the problem of decreasing likelihoods, [2] comes at the cost of training an external reward model, unlike other regularization methods. We will include further discussion around this related method. Regarding DPO-Positive [3], we agree this would be a valuable comparison: as it requires retraining from scratch, we may not be able to complete experiments within the rebuttal period, but we will aim to include it in the camera-ready version.
>
> **Lambda sensitivity**
>
> With the ablation in Sec5.2 we addressed the most pressing question of whether the regularization itself would provide benefits beyond length normalization alone. Nonetheless, the reviewer raises a valid point that including the full sensitivity analysis would help draw a clearer picture of the method robustness to lambda. The full sensitivity results are reported below, and will be included in the revised paper.
>
> *Reward Accuracy:*
>
> | | │ | Llama-3.1-8B | | │ | Mistral-7B | | │ | OLMo-7B | |
> |:---:|:---:|:---:|:---:|:---:|:---:|:---:|:---:|:---:|:---:|
> | **λ** | │ | **N-DPO** | **N-SimPO** | │ | **N-DPO** | **N-SimPO** | │ | **N-DPO** | **N-SimPO** |
> | 0.025 | │ | 79.05 | 78.59 | │ | 84.61 | 82.18 | │ | 75.98 | 65.22 |
> | 0.05 | │ | 79.17 | 78.94 | │ | 84.20 | 82.06 | │ | 75.87 | 63.25 |
> | 0.1 | │ | 78.94 | 79.34 | │ | 83.28 | 81.77 | │ | 76.16 | 62.62 |
>
> *Reward Margins:*
>
> | | │ | Llama-3.1-8B | | │ | Mistral-7B | | │ | OLMo-7B | |
> |:---:|:---:|:---:|:---:|:---:|:---:|:---:|:---:|:---:|:---:|
> | **λ** | │ | **N-DPO** | **N-SimPO** | │ | **N-DPO** | **N-SimPO** | │ | **N-DPO** | **N-SimPO** |
> | 0.025 | │ | 3.301 | 6.143 | │ | 2.355 | 7.223 | │ | 2.588 | 0.318 |
> | 0.05 | │ | 3.221 | 5.784 | │ | 2.238 | 7.084 | │ | 2.444 | 0.268 |
> | 0.1 | │ | 3.050 | 5.439 | │ | 2.092 | 6.856 | │ | 2.299 | 0.213 |
>
> Overall, N-DPO reward accuracy is stable across all lambdas and models, while N-SimPO is robust on Llama and Mistral but degrades on OLMo at higher lambda values, consistent with the delicate optimization landscape of this model (see our response to Reviewer on7F). Reward margins decrease monotonically with lambda for both methods (as expected: stronger regularization limits how far the model can separate chosen from rejected responses). Importantly, even the largest lambda (0.1) preserves healthy margins for N-DPO across all models, suggesting the method is not overly sensitive within the tested range.
>
> **Missing citations**
>
> We thank the reviewer for pointing out the relevant missing literature, and we will include the suggested citations in the revised paper.
>
> **Improving Fig2**
>
> We agree with the reviewer and thank them for this suggestion, which will be included in the final version. See also our response to Reviewer JH42.

---

> > ### Author Rebuttal · Reviewer_C1x4 · 2026-04-03
> >
> > Thanks for the response. The rebuttal does address my questions, about 1) confidence intervals (Please include them in the main tables as mean +/- stdev in the updated version), 2) Sensitivity to lambda. But some of my questions were unresolved
> >
> > 1) Baseline Comparisons. I agree that rerunning the entire suite of experiments with the additional baselines, in the rebuttal time window, is infeasible. But it would greatly help if their proposed regularization approach were compared to already established baselines (listed in my review). A comparison with DPO Positive on just one model (eg: llama) and one domain would greatly help in positioning their proposed regularization approach and quell any of my doubts
> >
> > 2) The generation case study was to observe if the proposed regularization leads to observed differences in generations compared to baseline, on account of the comment on decreasing likelihood of outlier tokens, in the paper.
> >
> > 3) Why does the reward accuracy increase with $\lambda$ for LLama N-SimPO, while other settings show a decreasing trend?
> >
> > I will be maintaining my score, for now.

---

> > > ### Author Response · Authors · 2026-04-08
> > >
> > > ## 1 - Baseline comparisons
> > >
> > > As requested, we managed to complete training and evaluation for both the DPOP [3] and DPO+SFT [Liu et al., 2024] baselines. Updated results are reported in the response to Reviewer 2jMo.
> > >
> > > **DPO-Positive** We conducted a hyperparameter sweep mimicking the setup from our previous experiments (particularly, for LR and $\beta$). To ensure a fair comparison, for the DPOP penalty coefficient $\lambda$, we included both values from the range originally proposed by the authors ($\lambda\in[5,500]$) and $\lambda=0.5$ (which is closer to our regularization scale). We found that smaller $\lambda$ values provide the best results, suggesting DPOP might benefit from further tuning; we will explore this more in detail in the final version.
> > >
> > > **DPO+SFT** We extend the results available in the paper for the DPO+SFT-trained Mistral and OLMo models, by including their evaluations on LM-eval-harness.
> > >
> > > **Discussion** The comparison reveals a clear distinction between approaches that address likelihood displacement _directly_ (N-DPO, through reward normalization) versus _indirectly_ (DPOP and DPO+SFT, through auxiliary penalties or objectives):
> > >
> > > - *Generation quality (AE1):*  N-DPO achieves competitive win rates across all models, and the highest on OLMo (81.24). DPOP is competitive on Mistral/Llama but substantially weaker on OLMo (69.03). DPO+SFT is strong on Llama (93.52) but weaker on Mistral (92.15) and OLMo (66.50)
> > >
> > > - *Common sense benchmarks (LMEH):* DPO+SFT achieves the best benchmark preservation on Mistral (+3.88 avg), and DPOP also performs well (+0.81). However, this comes at a significant cost to reward modeling quality (see below). On OLMo, DPO+SFT degrades substantially (-4.89), while both DPOP (+0.13) and N-DPO (+0.72) maintain or improve scores – with N-DPO providing the larger improvement. On Llama, N-DPO (+3.04) outperforms DPOP (+1.38).
> > >
> > > - *Reward accuracy:* This is where the distinction is sharpest. N-DPO achieves the strongest reward accuracy across models and datasets. Both DPOP and DPO+SFT show substantially weaker reward modeling: DPO+SFT on Mistral UF drops to 53.18 (near random) despite its strong LMEH scores.
> > >
> > > This pattern is consistent with the paper's central thesis: DPO+SFT and DPOP mitigate likelihood displacement through auxiliary objectives that preserve benchmark performance, but do not address the underlying lack of reward normalization. As a result, their implicit reward models remain poorly calibrated. N-DPO, by directly normalizing the reward, achieves a more favorable trade-off across generation quality, benchmark preservation, and reward modelling.
> > >
> > > &nbsp;
> > >
> > > ## 2- Generation case-study
> > >
> > > Following the reviewer's request, we conducted a token-level analysis comparing outlier (per-token reward values below 5th percentile) token distributions in DPO vs N-DPO and SimPO vs N-SimPO. We focused on OLMo-7B as it is the model where N-SimPO showed the most unstable downstream performance (see our response to Reviewer on7F), making it the most informative test case for whether the regularizer is operating as intended at the token level.
> > >
> > >
> > > Across all methods, the most frequent outlier tokens are consistently semantically unimportant. Newlines (197 occurrences), EOT (37), periods (24), and function words (such as the (13), is (11), and (10)) dominate. Tokens appearing exclusively in one method are rare subword fragments with counts of 1-3. This confirms that outlier tokens are overwhelmingly non-content-bearing, and their core set stable across methods.
> > >
> > > Our regularizer substantially reduces outlier severity. For N-DPO, the most extreme outlier reward improves from -57.39 to -27.89, and its mean from -5.90 to -4.55. The proportion of whitespace among outliers drops from 19.7% to 9.0%, in favor of content words, suggesting that indeed the regularizer prevents non-content-bearing tokens from being suppressed. For N-SimPO the effect is even cleaner: outlier mean improves from -4.13 to -3.23, extreme minimum from -15.06 to -9.67, reward std decreases from 0.52 to 0.45, and whitespace outliers drops from 31.6% to 5.2%.
> > >
> > > These results provide direct evidence that the regularization mitigates the extreme token-level reward displacements discussed in Sec4, consistent with the theoretical guarantees provided by the gradient analysis in Eq16.
> > >
> > > &nbsp;
> > >
> > > ## 3 - Llama N-SimPO reward accuracy increasing with $\lambda$
> > >
> > > We thank the reviewer for this careful observation. The sensitivity results refer to the best configuration (highest AE1 score) from a _joint_ sweep over both $\lambda$ and other hyperparameters (including $\beta$), so the reported results do not stem from varying $\lambda$ in isolation. The unexpected behavior of the reward accuracy is consistent with our broader observation that the optimization landscape for SimPO-based methods involves more complex hyperparameter interactions than DPO-based ones (see also response to Reviewer on7F).

---

### Official Review · Reviewer_2jMo · 2026-03-13

**Soundness:** 3
**Presentation:** 3
**Significance:** 2
**Originality:** 2
**Overall Recommendation:** 4
**Confidence:** 3

**Summary:**

This paper proposes a regularization term that can be directly incorporated into the loss functions of Direct Alignment Algorithms (DAAs) to address the over-optimization problem. Through empirical comparison of token-wise reward distributions and theoretical gradient analysis, the authors demonstrate the regularization term's effectiveness in handling outlier tokens. Experiments primarily validate the approach on DPO and SimPO, and comparisons against baselines on Instruction Following and Common Sense & Reasoning benchmarks reveal better trade-offs between generation quality and benchmark performance. The authors also evaluate implicit reward modeling capabilities, showing that N-SimPO and N-DPO achieve improved generalization.

**Compliance With Llm Reviewing Policy:**

Affirmed.

**Final Justification:**

I am raising my score from 3 to 4. The rebuttal addressed my main concerns satisfactorily.

My primary concern was the missing N-DPO vs. DPO+SFT comparison on reasoning benchmarks. The authors have committed to including this, and their clarification that their regularizer targets the root cause of likelihood displacement — probability mass redistribution to unseen responses — rather than merely the symptom of likelihood decrease, is a convincing and important distinction that should be made explicit in the main text.

The mixed results across settings remain a limitation, though the authors' explanation of the OLMo-7B case is plausible, and the consistent improvements of N-DPO across all three models on reasoning benchmarks are reassuring. The authors' candid acknowledgment of limitations in online RL and KTO settings is appropriate.

The promised revisions — expanded benchmark comparisons, clarified mathematical definitions, revised Figure 2, and explicit discussion of applicability limits — should meaningfully strengthen the final paper.

**Key Questions For Authors:**

1. In Appendix A., the AlpacaEval scores for the DPO+SFT are provided. How does N-DPO compare to DPO+SFT on common reasoning benchmarks? This comparison is important for understanding whether the proposed regularization offers advantages beyond what a simple SFT regularizer already provides, and the answer would meaningfully affect the evaluation of the paper's contribution.

**Limitations:**

yes

**Strengths And Weaknesses:**

- Shoundness

  The submission is technically sound overall. The key claims are supported as follows:

  1. *Likelihood displacement stems from the inability to account for unseen response distributions*: The authors justify this via a minimal illustrative example with three output options, comparing the limited training responses against the full response space. While intuitive, this justification is somewhat weak — more rigorous empirical or theoretical analysis would substantially strengthen the argument.
  2. *Outlier tokens are primary contributors to likelihood displacement*. Well supported by Figure 3.
  3. *The regularization mitigates outlier token impact by leveraging low-likelihood tokens*. Well supported by the gradient analysis.
  4. *The regularization improves trade-offs between generation quality and benchmark performance*. Validated across both LLM-as-a-Judge and common sense & reasoning benchmarks.

  The methods follow a standard preference alignment pipeline with standard evaluation benchmarks, and the theoretical proofs appear correct under reasonable assumptions. The authors are candid about both performance gains and drops in their evaluation.

- Presentation

  The paper is clearly written and well-structured, with a narrative that is easy to follow. It appropriately situates itself within the prior literature on RLHF, DPO, and SimPO. A few minor issues are worth noting:

  1. Line 059, left column: "without" may should be "with."
  2. Figure 1: The likelihood of C after the DPO step should be 0.7 (increasing), not as currently shown.

  Clarity would further benefit from explicit mathematical definitions of:

  1. Overall token-wise reward. At line 208, right column.

  2. The minimum token-wise reward per sample. At line 210, right column.
  3. The total length-normalized response probabilities $P_\theta(x), P_\text{ref}(x)$



- Significance

  The paper addresses a well-recognized and important problem — likelihood displacement is a common failure mode in preference alignment. However, the significance has some limitations:

  - While the paper identifies outlier tokens as primary drivers of likelihood displacement, this observation is arguably intuitive rather than surprising. Moreover, the problem is not fully resolved: Figure 2 shows that N-SimPO exhibits *more severe* likelihood displacement than SimPO in some settings.
  - Experimental results are mixed — N-DPO and N-SimPO outperform baselines in some settings but underperform in others, making it difficult to draw strong conclusions about consistent gains.
  - The proposed regularization is specialized to the standard DAA pipeline, requiring a fixed $(x, y_w, y_l)$ input format and a reference score $\pi_\text{ref}$, even when the base algorithm is reference-free (e.g., SimPO). This limits its applicability to broader machine learning settings.
  - That said, if the authors can demonstrate consistent improvements across a wider range of DAAs, this regularization could form the basis of a practical, reusable library for preference optimization.



- Originality

  The paper offers genuine novelty and insight:

  - It provides a new perspective on likelihood displacement, framing it as a normalization problem.
  - The proposed regularization term appears novel, and the motivation for combining it with both reference-based and reference-free DAAs is well-articulated.
  - The token-wise gradient analysis sheds new light on *why* DAAs can improve generation quality despite decreasing chosen response likelihoods.
  - One gap in the originality discussion is the absence of a comparison to existing regularization approaches such as DPO+SFT (where the SFT loss can be understood as serving a regularization function). This connection deserves explicit discussion to properly distinguish the contribution.

---

> ### Author Rebuttal · Authors · 2026-03-31
>
> **Comparison with DPO+SFT and other regularizations**
>
> This is a relevant point raised by multiple reviewers (see also C1x4, on7F), and we agree that more detailed results would help draw a clearer picture of the validity of our method. As requested, we will expand the DPO+SFT comparison to include common sense and reasoning benchmarks in the revised paper.
>
> More generally, regarding the difference between our proposed method and other SFT-based regularizations, we believe there is a meaningful distinction: methods like DPO+SFT and DPO-P address the *symptom* of response likelihood decrease (DPO+SFT does so unconditionally via an NLL loss, while DPO-P applies a penalty when likelihoods drop below those of the reference model). They do not however prevent the redistribution of probability mass to unseen responses. By encouraging the preservation of the total probability mass over response pairs, our regularizer targets instead the *root cause* of the likelihood displacement phenomenon. We will add an explicit discussion of this distinction in the revised paper.
>
> **Wrong likelihoods in example**
>
> We respectfully clarify that Fig1 is correct as intended. The key distinction is between *likelihoods* (unnormalized scores) and *probabilities* (normalized). In the example, DPO decreases the likelihoods of both A and B (which it observes during training), while the likelihood of C is unchanged (since DPO never sees C). However, since the total likelihood (i.e., the partition function) has decreased, the *probability* of C increases spuriously after normalization. This is precisely the phenomenon our regularizer addresses: by conserving total probability mass, we aim to prevent such spurious redistribution. We will add a clarifying note to the caption to make the likelihood/probability distinction more explicit.
>
> **Missing definitions**
>
> We thank the reviewer for flagging this. Below, we provide the required explicit definitions, which will be included in the main text
>
> - For each token $y^{(i)}$ in a response $y$ to a prompt $x$, the reward for that token is $\log \frac{\pi_\theta(y^{(i)} | x)}{\pi_\text{ref}(y^{(i)} | x)}$, and the sum of the token-wise rewards is the response’s reward
>
> - The minimum token-wise reward per sample is $\min_{i \in [L]} \left( \log \frac{\pi_\theta(y^{(i)} | x)}{\pi_\text{ref}(y^{(i)} | x)} \right)$, where $L$ is the length of $y$.
>
> - The length-normalized log-probability of a response $y$ with length $L$ to a prompt $x$ for a model $\pi$ is $\frac{1}{L} \log \pi(y | x)$
>
>
> **Limited applicability to broader settings**
>
> We thank the reviewer for highlighting this limitation. We note that for offline pairwise methods like SimPO, the practical impact is modest, as reference model probabilities can be precomputed once and stored alongside the training data: this prevents having to interrogate the original model at training, and adds only a negligible overhead. However, for other paradigms the reviewer's concern is absolutely well-founded: in online RL settings, responses are generated on-the-fly and reference probabilities cannot be precomputed, while methods like KTO operate on unpaired responses, meaning the formulation of our regularizer (built around matched chosen/rejected pairs) does not directly apply. We will flag these limitations explicitly in the revised paper, and we thank the reviewer for pointing towards this interesting direction for future work.
>
>
> **Mixed results across settings**
>
> We acknowledge that results are not uniformly positive across all model-method combinations. However, we emphasize that N-DPO consistently improves over DPO on reasoning benchmarks across all three models, and on generation quality as measured by AE2 (the held-out test metric). For N-SimPO, the picture is more nuanced: the main case where it underperforms is OLMo-7B-SFT on reasoning benchmarks. As we discuss in the paper, SimPO on OLMo already required very conservative hyperparameters (smallest learning rate, margin, and beta among all models), suggesting a delicate optimization landscape. N-SimPO appears to enable more aggressive preference optimization on this model, which improves generation quality but comes at the cost of reasoning benchmark performance. We believe the consistent improvements for N-DPO and the overall trend of better generation-benchmark trade-offs support the value of the approach.
>
> **N-SimPO displacement in Fig2**
>
> We agree the figure makes likelihood displacements hard to parse: we will revise it with aligned y-scales and combined chosen/rejected plots (see response to JH42). Regarding the interpretation: rather than eliminating displacement, N-SimPO achieves a larger reward margin (wider gap between chosen and rejected likelihoods) with a similar degree of displacement, which we believe explains the improved downstream performance. We acknowledge the effect is less clean than for N-DPO, where displacement is clearly mitigated.

---

> > ### Author Rebuttal · Reviewer_2jMo · 2026-04-03
> >
> > Thanks for the responses. I have updated my score.

---

> > > ### Author Response · Authors · 2026-04-08
> > >
> > > We thank the reviewer for the constructive engagement and updated score. As promised, we provide below the full comparison including DPOP and DPO+SFT on all evaluation axes. We refer to our response to Reviewer C1x4 for a detailed discussion of these results. In summary, while DPO+SFT and DPOP can preserve benchmark performance through auxiliary objectives, they do so without addressing the underlying reward normalization – resulting in substantially weaker implicit reward models (eg, DPO+SFT on Mistral UF: 53.18, near random). Our N-DPO approach, on the other hand, achieves competitive or superior results across all the considered evaluation axes, confirming that targeting the root cause of likelihood displacement yields a more favorable trade-off overall.
> > >
> > > &nbsp;
> > >
> > > ### Table 1: AlpacaEval1 Win Rate
> > >
> > >
> > > ||Mistral-7B|Llama-3.1-8B|OLMo-7B-SFT|
> > > |--|--|--|--|
> > > |Ref|93.17|90.00|58.15|
> > > |DPO|94.66|91.67|79.63|
> > > |N-DPO|94.28|91.28|81.24|
> > > |SimPO|90.87|88.93|71.21|
> > > |N-SimPO|92.72|85.45|79.63|
> > > |DPO+SFT|92.15|93.52|66.50|
> > > |DPOP $\lambda=0.5$|94.52|91.28|69.03|
> > > |DPOP $\lambda=5$|94.27|91.04|66.13|
> > >
> > > &nbsp;
> > >
> > > ### Table 2: LM-EvalHarness, deltas from ref
> > >
> > >
> > > #### Mistral-7B-Instruct-v0.2
> > >
> > >
> > > ||MMLU|ARC-C|ARC-E|HellaS|PiQA|SciQ|WinoG|Avg|
> > > |--|--|--|--|--|--|--|--|--|
> > > |Ref|44.32|49.49|70.33|62.86|75.19|90.90|61.48|64.94|
> > > |DPO|-1.34|-3.41|-5.05|-1.45|-2.67|-3.00|-2.68|-2.80|
> > > |N-DPO|+0.17|-1.11|-1.90|-0.95|-1.96|-0.80|-0.31|-0.98|
> > > |SimPO|-0.70|-3.84|-3.12|-7.47|-5.06|-0.50|-2.21|-3.27|
> > > |N-SimPO|-0.84|-4.10|-3.62|-6.87|-4.79|-1.20|-2.44|-3.41|
> > > |DPO+SFT|+2.57|+6.14|+4.75|+2.34|+3.37|+1.60|+6.40|+3.88|
> > > |DPOP $\lambda=0.5$|+1.65|+0.25|+0.80|+1.38|+0.44|+0.30|+0.87|+0.81|
> > > |DPOP $\lambda=5$|+0.36|-0.75|-0.20|+0.28|+0.11|+0.20|+0.08|+0.01|
> > >
> > >
> > > #### Llama-3.1-8B-Instruct
> > >
> > >
> > > ||MMLU|ARC-C|ARC-E|HellaS|PiQA|SciQ|WinoG|Avg|
> > > |--|--|--|--|--|--|--|--|--|
> > > |Ref|43.28|52.82|81.44|57.52|79.76|95.20|67.40|68.20|
> > > |DPO|+4.71|-4.78|-9.51|+6.43|-5.82|-4.20|-9.55|-3.25|
> > > |N-DPO|+4.24|+3.84|+0.80|+6.24|+0.82|+0.20|+5.13|+3.04|
> > > |SimPO|+1.69|+4.77|+0.17|-5.49|-0.54|+0.80|+6.55|+1.14|
> > > |N-SimPO|+8.49|+9.21|+2.27|-0.92|+1.36|+1.10|+4.33|+3.69|
> > > |DPO+SFT|-|-|-|-|-|-|-|-|
> > > |DPOP $\lambda=0.5$|+7.08|+0.41|+0.59|+0.36|+0.22|+0.00|+1.03|+1.38|
> > > |DPOP $\lambda=5$|+7.20|+0.68|+0.18|+0.75|+0.71|+0.30|+1.69|+1.64|
> > >
> > >
> > > #### OLMo-7B-SFT
> > >
> > >
> > > ||MMLU|ARC-C|ARC-E|HellaS|PiQA|SciQ|WinoG|Avg|
> > > |--|--|--|--|--|--|--|--|--|
> > > |Ref|37.94|39.33|67.97|53.54|76.66|91.10|63.93|61.50|
> > > |DPO|-0.15|-0.85|-2.06|+0.27|-1.42|-4.80|-2.05|-1.58|
> > > |N-DPO|+0.47|+1.88|+2.11|+3.52|-0.82|-1.20|-0.95|+0.72|
> > > |SimPO|-0.04|-0.34|-0.25|-1.17|-1.09|-0.70|-1.50|-0.73|
> > > |N-SimPO|+0.10|-1.53|-14.01|-3.49|-4.79|-4.20|-5.76|-4.81|
> > > |DPO+SFT|-0.62|-4.18|-8.21|-3.46|-0.49|-12.80|-4.50|-4.89|
> > > |DPOP $\lambda=0.5$|+0.84|-0.42|+2.11|-2.48|+1.01|-1.80|+1.63|+0.13|
> > > |DPOP $\lambda=5$|+0.74|-0.34|+2.52|-3.53|+0.85|-1.40|+1.47|+0.04|
> > >
> > > &nbsp;
> > >
> > > ### Table 3: Reward Acc
> > >
> > >
> > > ||Mistral-7B|||Llama-3.1-8B|||OLMo-7B-SFT|||
> > > |--|--|--|--|--|--|--|--|--|--|
> > > ||UF|HH|HS|UF|HH|HS|UF|HH|HS|
> > > |DPO|80.73|55.12|66.15|74.71|59.55|68.23|71.12|53.30|64.06|
> > > |N-DPO|94.28|81.94|56.33|77.20|58.73|75.00|73.26|54.63|64.32|
> > > |SimPO|82.12|57.17|64.32|76.22|57.03|67.19|60.19|55.47|53.65|
> > > |N-SimPO|82.06|56.82|63.28|75.81|57.62|67.97|65.05|55.68|56.77|
> > > |DPO+SFT|53.18|50.96|59.11|-|-|-|59.14|51.91|66.15|
> > > |DPOP $\lambda=0.5$|72.74|56.74|55.73|61.92|59.28|55.21|63.25|55.49|53.65|
> > > |DPOP $\lambda=5$|63.19|54.84|47.66|65.39|60.04|58.33|53.76|52.27|47.66|
> > >
> > > Unfortunately, the checkpoints for the Llama DPO+SFT model were corrupted, but we will re-train and include them in the final version. Moreover, we were unable to evaluate DPO-P on AE2 as the judge model we used was retired in the meantime. For consistency, we will re-run AE2 evals for all models considered with a new judge for the final version.

---

### Decision · Program_Chairs · 2026-04-30

**Decision:**

Accept (regular)

**Comment:**

The final scores are 4443, and I recommend acceptance. The main concerns were about missing stronger comparisons with more baselines, evaluation without standard errors, lack of hyperparam sensitivity study, among a few other concerns. The rebuttal addressed many of these concerns with more empirical results and comparison with baselines, and made promises to add more discussions, improve figures, expand related work, improve/clarify evaluation protocol.